# Comparative analysis reveals the long-term coevolutionary history of parvoviruses and vertebrates

**Matthew A. Campbell[1]⊗\*, Shannon Loncar[2]⊗¤, Robert M. Kotin[2,3], Robert J. Gifford[4]\***

**1** University of Alaska Museum of the North, Fishes and Marine Invertebrates, Fairbanks, Alaska, United States of America, **2** University of Massachusetts Medical School, Department of Microbiology and Physiological Systems, Gene Therapy Center, Worcester, Massachusetts, United States of America, **3** Carbon Biosciences, Lexington, Massachusetts, United States of America, **4** MRC-University of Glasgow Centre for Virus Research, Bearsden, Glasgow, United Kingdom

⊗ These authors contributed equally to this work.
¤ Current address: Carbon Biosciences, Lexington, Massachusetts, United States of America
\* robert.gifford@glasgow.ac.uk

**Data Availability Statement:** All data are available in Parvovirus-GLUE, and open repository and cross-platform database hosted https://zenodo.org/record/6968218

## Abstract

Parvoviruses (family *Parvoviridae*) are small DNA viruses that cause numerous diseases of medical, veterinary, and agricultural significance and have important applications in gene and anticancer therapy. DNA sequences derived from ancient parvoviruses are common in animal genomes and analysis of these *endogenous parvoviral elements* (EPVs) has demonstrated that the family, which includes twelve vertebrate-specific genera, arose in the distant evolutionary past. So far, however, such "paleovirological" analysis has only provided glimpses into the biology of ancient parvoviruses and their long-term evolutionary interactions with hosts. Here, we comprehensively map EPV diversity in 752 published vertebrate genomes, revealing defining aspects of ecology and evolution within individual parvovirus genera. We identify 364 distinct EPV sequences and show these represent approximately 200 unique germline incorporation events, involving at least five distinct parvovirus genera, which took place at points throughout the Cenozoic Era. We use the spatiotemporal and host range calibrations provided by these sequences to infer defining aspects of long-term evolution within individual parvovirus genera, including mammalian vicariance for genus *Protoparvovirus*, and interclass transmission for genus *Dependoparvovirus*. Moreover, our findings support a model of virus evolution in which the long-term cocirculation of multiple parvovirus genera in vertebrates reflects the adaptation of each viral genus to fill a distinct ecological niche. Our findings show that efforts to develop parvoviruses as therapeutic tools can be approached from a rational foundation based on comparative evolutionary analysis. To support this, we published our data in the form of an open, extensible, and cross-platform database designed to facilitate the wider utilisation of evolution-related domain knowledge in parvovirus research.

**Funding:** This work was supported by funding from the Association Monégasque Contre les Myopathies (RK), and the Bill & Melinda Gates Foundation (OPP1202116 to RK). The funders had no role in study design, data collection and analysis, decision to publish, or preparation of the manuscript.

**Competing interests:** I have read the journal's policy and the authors of this manuscript have the following competing interests: R.M.K. is a co-founder of Carbon Therapeutics, Inc., which is a co-assignee of a patent application filed on behalf of University of Massachusetts Medical School and Carbon Biosciences, Inc.

**Abbreviations:** aa, amino acid; AAV, adeno-associated virus; ArcPV, archeoprotoparvovirus; BLAST, Basic Local Alignment Search Tool; DIGS, Database-Integrated Genome Screening; EPV, endogenous parvoviral element; EVE, endogenous viral element; GLUE, Genes Linked by Underlying Evolution; HHV6, human herpesvirus 6; ICTV, International Committee for Taxonomy of Viruses; ITR, inverted terminal repeat; kb, kilobases; ML, maximum likelihood; MSA, multiple sequence alignment; Mya, million years ago; NCBI, National Center for Biotechnology Information; ORF, open reading frame; PLA2, phospholipase A2; WGS, whole genome sequencing.

## Introduction

Parvoviruses (family *Parvoviridae*) are a diverse group of small, nonenveloped DNA viruses that infect a broad range of animal species [1,2]. The family includes numerous important pathogens of humans and domesticated species, including erythroparvovirus B19 (fifth disease) [3], carnivore protoparvovirus 1 (canine parvovirus) [4], and carnivore amdoparvovirus 1 (Aleutian mink disease) [5]. Parvoviruses are also being developed as next-generation therapeutic tools: Adeno-associated virus (AAV) has been successfully adapted as a gene therapy vector, and other parvoviruses are leading candidates for human gene therapy [6,7]. Rodent protoparvoviruses show natural oncotropic and oncolytic properties and are being explored as potential anticancer therapeutics [8–10].

Parvoviruses have highly robust, icosahedral capsids (T = 1) that contain a linear, single-stranded DNA genome approximately 5 kilobases (kb) in length. Their compact genomes are typically organised into two major gene cassettes, one (Rep/NS) that encodes the nonstructural replication proteins, and another (Cap/VP) that encodes the structural coat proteins of the virion [11]. However, some genera contain additional open reading frames (ORFs) adjacent to these genes or overlapping them in alternative reading frames. All parvovirus genomes are flanked at their 3′ and 5′ ends by palindromic inverted terminal repeat (ITR) or "telomere" sequences that are the only *cis* elements required for replication.

In recent years, high-throughput sequencing and new metagenomic analytical methods have led to the discovery of numerous novel parvovirus species, and the taxonomy of the family *Parvoviridae* has now been extensively reorganised to accommodate this newly discovered diversity [1,2]. The availability of genome sequence data from a wide range of diverse parvovirus species provides unprecedented opportunities to utilise comparative approaches to investigate parvovirus biology. Furthermore, progress in whole genome sequencing (WGS) has revealed that DNA sequences derived from parvoviruses (and many other virus groups) are widespread within metazoan genomes [12–14]. Such "*endogenous viral elements*" (EVEs) arise when infection of germline cells results in virus-derived DNA sequences being incorporated into chromosomes and inherited as host alleles. EVE sequences can sometimes persist in the gene pool over many generations with the result that some are genetically "fixed" (i.e., they reach a frequency of 100%). Fixed EVEs have unique value to studies of virus evolution because—much like a virus "fossil record"—they preserve retrospective information from which the evolutionary interactions of viruses and hosts across geologic timescales can be inferred. For example, identification of orthologous EVE loci in multiple related host species demonstrates that virus integration occurred in the common ancestor of those species, prior to their divergence [12,13]. A robust minimum age estimate for EVE integration can therefore be inferred from host species divergence times (which are in part derived from fossil evidence).

Comparative studies have shown that *endogenous parvoviral element* (EPV) sequences occur frequently in vertebrate genomes, and many of these derive from germline incorporation events that occurred million years ago (Mya) [15–18]. In this study, we perform broad-scale comparative analysis of 752 published vertebrate genomes to recover 364 distinct EPV sequences representing at least 199 unique loci and involving at least five distinct parvovirus genera. Through broad-scale phylogenetic and genomic analysis—encompassing all known vertebrate EPVs and parvovirus species—we reveal the long-term evolutionary interactions between parvoviruses and their vertebrate hosts.

## Results

### Open resources for comparative genomic analysis of parvoviruses

To facilitate greater reproducibility and reusability in comparative genomic analyses, we previously developed GLUE (Genes Linked by Underlying Evolution), a bioinformatics software

**Table 1. Summary of MSA hierarchy constructed for the family *Parvoviridae*.**

| # | Scope/Name | Parent | Children | Constraining reference | Coverage[a] | Number of taxa[b] | | |
|---|------------|--------|----------|------------------------|-------------|-------------------|---|---|
| | | | | | | Viruses[c2.] | | EVEs |
| | **Family (root MSA)** | | | | | | | |
| 1 | *Parvoviridae* | none | 3 | CPV | NS (13%) | 3 | | 0 |
| | **Subfamily** | | | | | | | |
| 2 | *Parvovirinae* | *Parvoviridae* | 2 | CPV | NS (63%) | 13 | | **4** |
| 3 | *Hamaparvovirinae* | *Parvoviridae* | 2 | PPV7 | NS | 5 | | 0 |
| 4 | *Densoparvovirinae* | *Parvoviridae* | 0 | JcDNV | NS | 9 | (1) | 0 |
| | **Cross-genus** | | | | | | | |
| 5 | Boca-Ave | *Parvovirinae* | 2 | ChPV | Genome (70%) | 2 | | 0 |
| 6 | Amdo-Proto | *Parvovirinae* | 2 | CPV | Genome (77%) | 2 | | 0 |
| 7 | EDCT | *Parvovirinae* | 3 | HPV4 | Genome (57%) | 4 | | 0 |
| 8 | Chaphama-Icthama | *Hamaparvovirinae* | 2 | PPV7 | Genome | 2 | | 0 |
| | **Genus** | | | | | | | |
| 9 | *Aveparvovirus* | Boca-Ave | 0 | ChPV | Genome (88%) | 4 | | 0 |
| 10 | *Bocaparvovirus* | Boca-Ave | 0 | BPV | Genome (75%) | 16 | | 0 |
| 11 | *Erythroparvovirus* | EDCT | 0 | B19 | Genome (80%) | 8 | (2) | **2** |
| 12 | *Tetraparvovirus* | EDCT | 0 | HPV4 | Genome (80%) | 9 | | 0 |
| 13 | *Dependoparvovirus* | EDCT | 0 | AAV2 | Genome (84%) | 28 | | **81** |
| 14 | *Copiparvovirus* | EDCT | 0 | BPV2 | Genome (62%) | 7 | | 0 |
| 15 | *Amdoparvovirus* | Amdo-Proto | 0 | AMDV | Genome (85%) | 7 | | **6** |
| 16 | *Protoparvovirus* | Amdo-Proto | 0 | CPV | Genome (90%) | 20 | | **106** |
| 17 | *Chaphamaparvovirus* | *Hamaparvovirinae* | 0 | PPV7 | Genome (85%) | 14 | (4) | **0** |
| 18 | *Icthamaparvovirus* | *Hamaparvovirinae* | 0 | SyIPV | Genome (62%) | 1 | | **1** |

[a]Coverage relative to constraining reference.

[b]Note that linking alignments (i.e., those representing internal nodes within the alignment tree hierarchy) contain only the reference sequences for their "child" alignments.

[c]Numbers shown in brackets indicate proportion of viral taxa that are putatively exogenous viruses identified in the present study by screening whole genome sequence databases.

AMDV, Aleutian mink disease virus; BPV, bovine parvovirus; B19, human erythroparvovirus B19; ChPV, chicken parvovirus; CPV, canine parvovirus; EDCT, Erythro-Dependo-Copi-Tetra group; EVE, endogenous viral element; HPV4, human parvovirus 4; JcDNV, *Junonia coenia* densovirus; MSA, multiple sequence alignment; NS, non-structural gene; PPV7, porcine parvovirus 7; SyIPV, *Syngnathus scovelli* ichthamaparvovirus.

framework for the development and maintenance of "virus genome data resources" [19]. Here, we used the GLUE framework to create Parvovirus-GLUE [20], an openly accessible online resource for comparative analysis of parvovirus genomes (**S1 and S2 Figs**). Data items collated in Parvovirus-GLUE include the following: (i) a set of 135 reference genome sequences (**S1 Table**) each representing a distinct parvovirus species and linked to isolate-associated data (isolate name, time and place of sampling, host species); (ii) a standardized set of 51 parvovirus genome features (**S2 Table**); (iii) genome annotations specifying the coordinates of these genome features within reference genome sequences (**S3 Table**); and (iv) a set of multiple sequence alignments (MSAs) constructed to represent distinct taxonomic levels within the family *Parvoviridae* (**Table 1 and S3 Fig**).

The Parvovirus-GLUE project is built by using GLUE's native command layer to create a bespoke MySQL database that not only contains the data items associated with our analysis, but also maps the semantic links between them (e.g., the associations between specific sequences, genome features, and MSA segments) (**S1** and **S2 Figs**). Standardised, reproducible

comparative genomic analyses can then be implemented by using GLUE's command layer to coordinate interactions between the project database and bioinformatics software tools.

Parvovirus-GLUE aims to provide a platform through which researchers working in different areas of parvovirus genomics can benefit from one another's work. The project can be installed on all commonly used computing platforms and is also fully containerised via Docker [21]. In the interests of maintaining a lightweight, flexible approach, the published project contains only a single reference genome for each parvovirus species. However, it can readily be extended to allow in-depth analysis at the species level (a tutorial included with the published resource demonstrates how this can be done; [20]). Parvovirus-GLUE is hosted in an openly accessible online version control system (GitHub), providing a platform for its ongoing development by the research community, following practices established in the software industry (**S1C Fig**) [22]. To facilitate its use across a broad range of analysis contexts, the resource adheres to a "data-oriented programming" paradigm that directly addresses issues of reusability, complexity, and scale in the design of information systems [23].

## Comprehensive mapping of endogenous parvoviral elements in published vertebrate genomes

To identify EPV loci in published vertebrate genomes, we performed systematic, similarity search-based in silico screening (see **S4 Fig**) of WGS data representing 752 vertebrate species. This led to the recovery of a total of 595 EPV sequences (**Fig 1**), which we resolved into a set of 199 distinct orthologous loci via sequence comparisons (**Fig 2**). We identified flanking genes for EPV loci (**S4**–**S6 Tables**) and compiled the robust, orthology-based minimum age calibrations we obtained from EPVs to generate an overview of parvovirus and vertebrate interaction over the past 100 My (**Fig 3**).

EPVs were identified in all major groups of terrestrial vertebrates except agnathans, crocodiles, and amphibians (**Table 2**). Overall, however, they were found to occur significantly more frequently in mammalian WGS assemblies than in those of other vertebrate groups, based on a two-sample proportion test implemented in the R software package [24], as follows: Mammalia versus Sauria: (178 loci in 353 mammalian genomes, prop = 0.50 versus 16 loci in 200 saurian genomes, prop = 0.08; $p$-value = $2.4 \times 10^{-23}$); Mammalia versus Actinopterygii (3 loci, 175 genomes, prop = 0.02; $p$-value = $3.69 \times 10^{-28}$).

To taxonomically classify EPVs, we used a combination of sequence similarity-based comparisons and phylogenetic analysis. We found the vertebrate EPVs were predominantly derived from viruses similar to protoparvoviruses (genus *Protoparvovirus*) and dependoparvoviruses (genus *Dependoparvovirus*). Meanwhile, the *Amdo-, Erythro-* and *Ichthamaparvovirus* genera are also represented in the parvovirus "fossil record" (**Fig 1**). Meanwhile, the *Ave-, Boca-, Tetra-, Copi-,* and *Chaphamaparvovirus* genera—all of which infect vertebrates—are conspicuously absent.

We identified 121 protoparvovirus-related EPV sequences in mammals, which we estimate to represent at least 105 distinct germline incorporation events (**S5 Table**). Several genome-length elements were identified, and most elements spanned at least approximately 50% of the genome (**Fig 2**). We also identified 213 dependoparvovirus-related EPV sequences, which we estimate to represent at least 80 distinct germline incorporation events (**S4 Table**). Dependoparvovirus EPVs were identified in a broad range of vertebrate classes, including mammals, birds, and reptiles (**Table 2**). Relatively few genome-length or gene-length elements are found among dependoparvovirus-derived EPVs (**Figs 2 and** S11).

We identified the first reported examples of EPVs derived from genus *Erythroparvovirus* in the genomes of the Patagonian mara (*Dolichotis patagonum*)—a New World rodent—and the

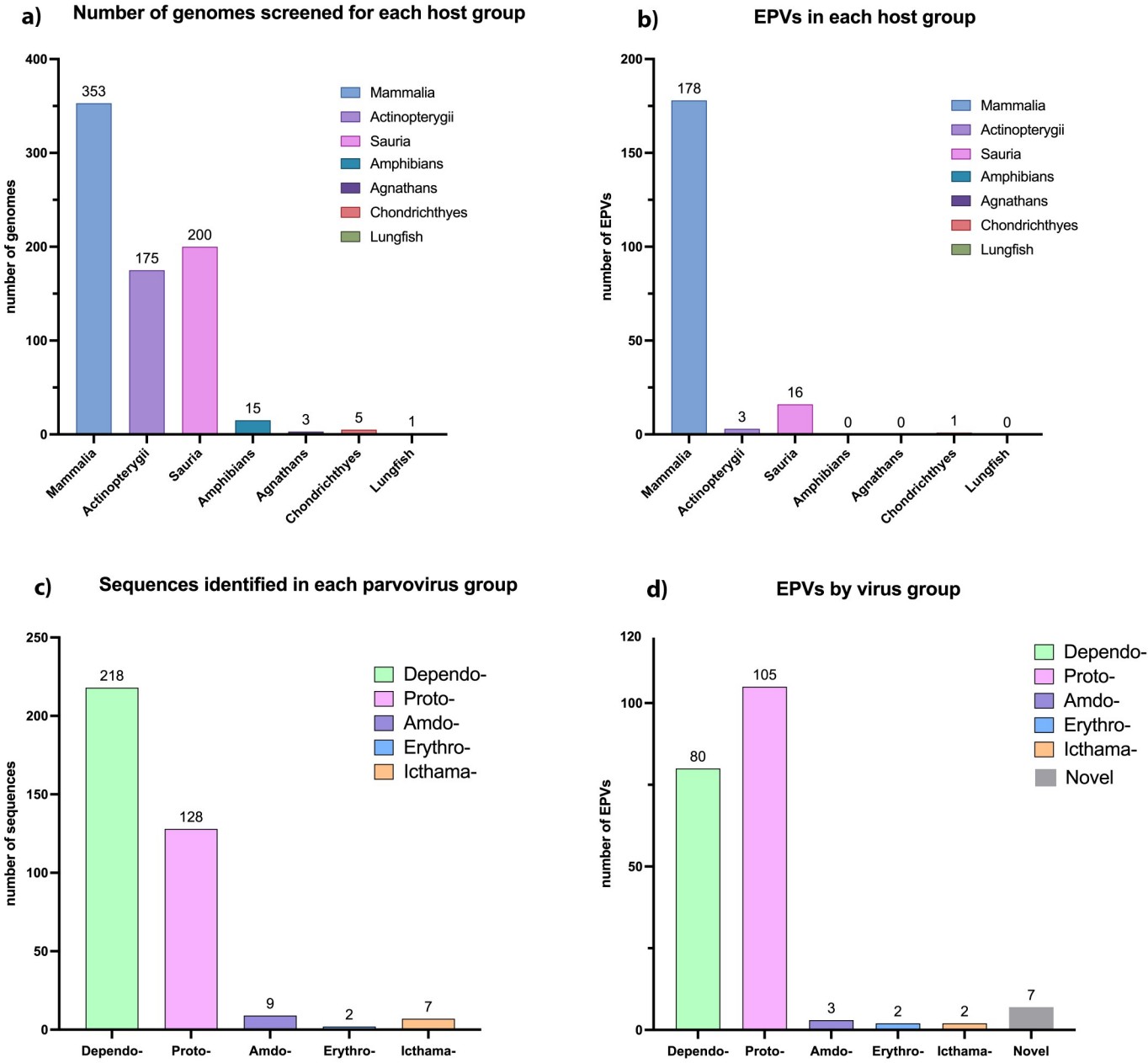

**Fig 1. Summary of EPV diversity identified via in silico screening. (a)** Number of genomes screened per host class. Sauria is comprised of birds (144 genomes) and reptiles (56 genomes). (b) Number of unique EPV loci identified in each host class. (c) Number of sequences identified in each parvovirus group. **(d)** Number of sequences identified in each parvovirus group. Graphs were plotted with GraphPad Prism9. The data underlying this figure can be found in https://zenodo.org/record/6968218#.Yu115vHMIUY.

Indri (*Indri indri*), a Malagasy primate (**Figs 2 and** S12). Amdoparvovirus-like EPVs have been reported previously [25]; however, our screen identified novel orthologous copies of *EPV-Amdo.101-Serpentes*, thereby providing a robust minimum age estimate of >100 Mya for this insertion and calibrating the evolutionary timeline of amdoparvoviruses (**Tables 3 and** S5).

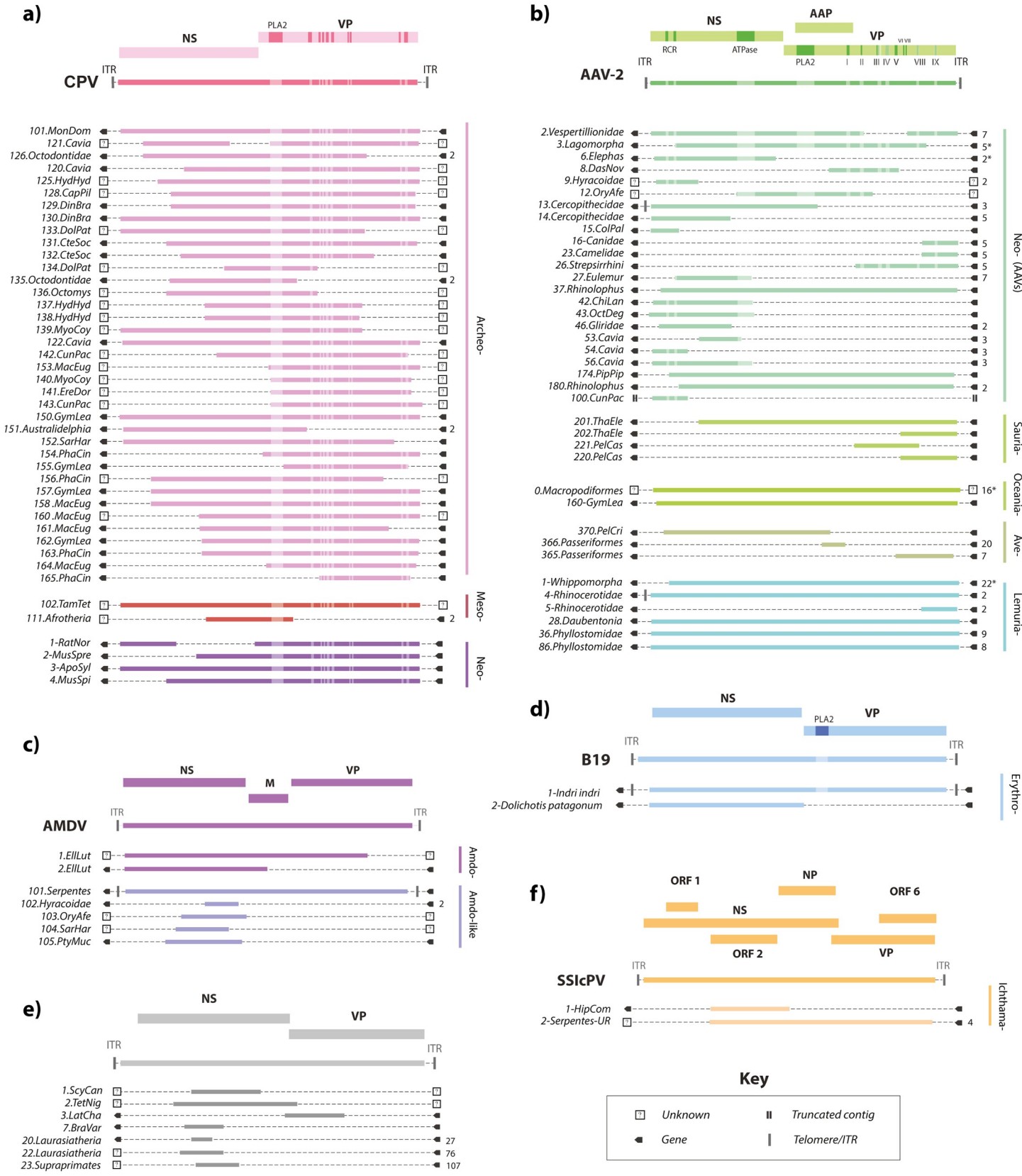

**Fig 2. Genomic structures of unique EPV loci. (a)** Protoparvovirus-derived EPV loci shown relative to the canine parvovirus (CPV) genome; **(b)** Dependoparvovirus-derived EPVs loci shown relative to the adeno-associated virus 2 (AAV-2) genome; **(c)** EPV loci derived from Amdoparvovirus-like viruses shown relative to the Aleutian mink disease (AMDV) genome; **(d)** Erythroparvovirus-derived loci shown relative to the parvovirus B19 genome; **(e)** EPVs derived from unclassified parvoviruses shown relative to a generic parvovirus genome. **(f)** Icthamaparvovirus-derived loci shown relative to *Syngnathus scovelli* parvovirus (SscPV). Solid bars to the right of each EPV set show taxonomic ranks below genus level. Numbers shown to the immediate right indicate a consensus and the number of orthologs used to create it. Asterisks indicate where this number includes sequences obtained in previous studies. Boxes bounding EPV elements indicate either (i) the presence of an identified gene (see **S4–S6 Tables**); (ii) an uncharacterised genomic flanking region; or (iii) a truncated contig sequence (see key). EPV locus identifiers are shown on the left. EPV were assigned unique identifiers (IDs) constructed from three components following a convention proposed for endogenous retroviruses [61]. The first component is the classifier "EPV." The second component comprises the name of the lowest level taxonomic group (i.e., species, genus, subfamily, or other clade) into which the element can be confidently placed by phylogenetic analysis and a numeric ID that uniquely identifies the insertion, separated by a period. The third component specifies the group of species in which the sequence is found. Six letter abbreviations are used here to indicate host species. **Genome feature abbreviations**: NS, nonstructural protein; VP, capsid protein; ORF, open reading frame; ITR, inverted terminal repeat; PLA2, phospholipase A2 motif. **Species name abbreviations:** PhaCin, Phascolarctos cinereus; GymLea, Gymnobelideus leadbeateri; SarHar, Sarcophilus harrisii; MacEug, Macropus eugenii; VomUrs, Vombatus ursinus; MonDom, Monodelphis domestica; OryAfe, Orycteropus afer; ChrAsi, Chrysochloris asiatica; ProCap, Procavia capensis; HetMeg, Heterohyrax brucei; EchTel, Echinops telfairi; TamTet, Tamandua tetradactyla; BraVar, Bradypus variegatus; DasNov, Dasypus novemcinctus; MegLyr, Megaderma_lyra; PipPip, Pipistrellus pipistrellus; EllLut, Ellobius lutescens; PedCap, Pedetes capensis; RatNor, Rattus norvegicus; MusSpr, Mus spretus; MusSpi, Mus spicelagus; ApoSyl, Apodemus sylvaticus; CapPil, Capromys pilorides; OctMim, Octomys mimax; CteSoc, Ctenomys sociabilis; EreDor, Erethizon dorsatum; GraMur, Graphiurus murinus; NanGal, Nannospalax galili; CunPac, Cuniculus paca; HydHyd, Hydrochoerus hydrochaeris; MyoCoy, Myocaster coypus; DinBra, Dinomys branickii; CasCan, Castor_canadensis; MusAve, Muscardinus avellanarius; ApoSyl, Apodemus sylvaticus; CraTho, Craseonycteris thonglongyai; OctDeg, Octodon degus; ChiLan, Chinchilla lanigera; CunPac, Cuniculus paca; GliGli, Glis glis; DolPat, Dolichotis patagonum; DauMad, Daubentonia madagascariensis; IndInd, Indri indri; ColAng, Colobus angolensis; ThaEle, Thamnophis elegans; PelCas, Pelusios castaneus; PelCri, Pelecanus crispus; EgrGar, Egretta garzetta; GuaGua, Guaruba guarouba; OpiHoa, Opisthocomus hoazin; HipCom, Hippocampus comes; PtyMuc, Ptyas mucosa; ScyCan, Scyliorhinus canicular; TetNig, Tetraodon nigroviridis. The data underlying this figure can be found in https://zenodo.org/record/6968218#.Yu115vHMIUY.

Subfamily *Hamaparvovirinae* contains two genera known to infect vertebrates—*Chaphamaparvovirus* and *Ichthamaparvovirus* [1]. We previously reported an *Ichthamaparvovirus*-derived EPV locus in fish [26]. Here, we report an additional locus in snakes (suborder Serpentes). This sequence demonstrates that *Ichthamaparvovirus* host range extends to reptiles (**Figs 2** and S13) and, via orthology across multiple snake species, establishes a minimum age of 62 My for the genus (**Table 3**).

## Phylogenetic analysis reveals the evolutionary history of subfamily Parvovirinae

Via Parvovirus-GLUE, we implemented a reproducible and extensible process (**S6 Fig**) for reconstructing evolutionary relationships across the entire *Parvoviridae*, at a range of taxonomic levels. Phylogenies were reconstructed using maximum likelihood (ML), firstly among viruses only (**S7 Fig**), and secondly among both viruses and EPVs (**Figs 4–8** and S8-S13). For subfamily *Parvovirinae*, we reconstructed phylogenies from polypeptide-level MSAs spanning the highly conserved tripartite helicase domain of Rep (**Fig 4**). These phylogenies reveal three robustly supported sublineages each encompassing multiple genera as follows: (i) "ETDC": *Erythro-*, *Tetra-*, *Dependo-*, and *Copiparvovirus*; (ii) "Ave-Boca": *Ave-* and *Bocaparvovirus*; and (iii) "Amdo-Proto": *Amdo-* and *Protoparvovirus*.

The EDTC and "Amdo-Proto" clades are demonstrably ancient as they both include EPVs that were incorporated into the germline >80 Mya. The "Ave-Boca" lineage does not have fossil representatives, but, notably, it comprises entirely distinct mammalian and saurian lineages, raising the possibility of ancient host–virus codivergence along the Mammalia-Sauria split approximately 200 Mya (**Table 3** and **Fig 3**). Similarly, we identified EPVs derived from the "Amdo-Proto" and "ETDC" lineages in basal vertebrates including lobe-finned fish (class Sarcopterygii) and sharks (class Chondrichthyes) (**Fig 4**). Consistent with ancient codivergence (rather than recent, interclass transmission), these sequences group basally, suggesting that the emergence of *Parvovirinae* genera might predate the deeper divergences among terrestrial vertebrates (**Table 3** and **Fig 3**).

While the majority of EPV loci identified in our study are unambiguously related to contemporary parvoviruses, several could not be classified beyond the subfamily level (all derive

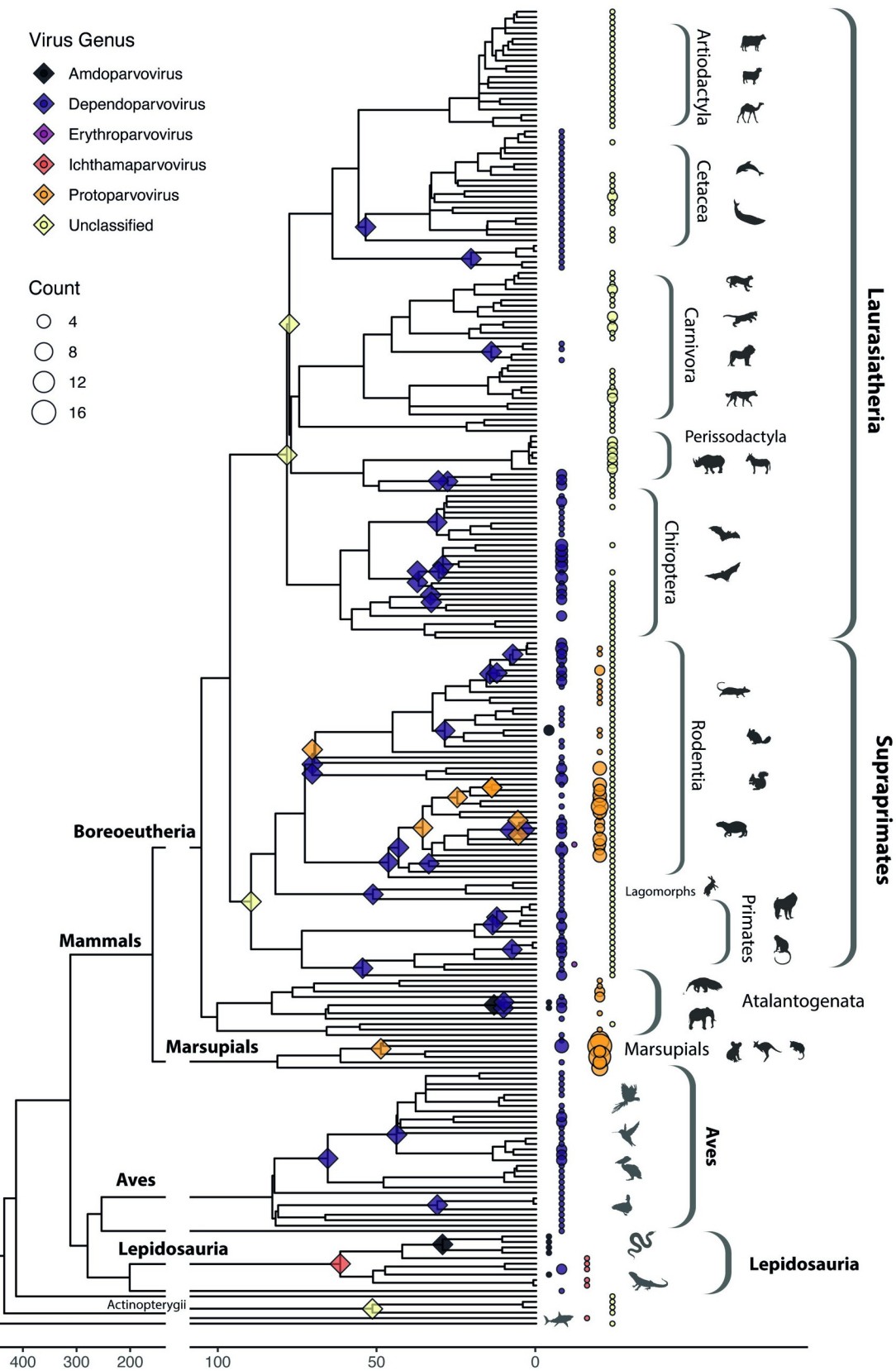

**Fig 3. Incorporation of EPVs into the vertebrate germline.** A time-calibrated evolutionary tree of vertebrate species examined in this study, illustrating the distribution of germline incorporation events over time. Colours indicate parvovirus genera as shown in the key. Diamonds on internal nodes indicate minimum age estimates for EPV loci endogenization (calculated for EPV loci found in >1 host species). Coloured circles adjacent to tree tips indicate the presence of EPVs in host taxa, with the diameter of the circle reflecting the number of EPVs identified (see count key). Brackets show taxonomic groups within vertebrates. The phylogeny shown here was obtained from TimeTree, a database of organism timelines, timetrees, and divergence times [35]. The data underlying this figure can be found in https://github.com/MacCampbell/parvoviridae-coevolution.

from viruses in subfamily *Parvovirinae*) (**S6 Table**). Some were very short and ancient (i.e., >80 Mya) and hence difficult to classify using phylogenetic approaches. These included a short VP-derived insert previously reported in the *limbin* gene locus of mammals belonging to superorder Euarchontoglires (Supraprimates) [15], and two short Rep-derived elements found in mammals belonging to superorder Laurasiatheria. Longer yet still unclassifiable EPVs were identified in lower vertebrate groups (e.g., lobe-finned fish). These EPVs might derive from members of extant parvovirus groups that are yet to be described.

Some herpesvirus (family *Herpesviridae*) lineages contain a homolog of the parvovirus *rep* gene in their genomes—called "U94" in human herpesvirus 6 (HHV6). This sequence—which is presumed to have arisen via parvovirus integration into an ancestral herpesvirus genome—groups in a nonspecific position within the *Parvovirinae* clade (**Fig 4**). U94 homologs occur in multiple members of genus *Roseolovirus (*subfamily *Betaherpesvirinae*) [27], suggesting insertion occurred following the divergence of Herpesviridae subfamilies approximately 200 Mya [28] (**Table 3**).

Viruses belonging to the 'Amdo-Proto' lineage have only been isolated from mammals, suggesting that both amdo- and protoparvoviruses might have originated in this host class, perhaps even relatively recently (e.g., within the past 20 My). However, the presence of a basal, ancient, amdoparvovirus-derived EPV (*Amdo.101-Serpentes*) in a squamate reptile (S3B Fig) suggests a more distant evolutionary separation between these groups (**Table 3**). Previous studies had suggested that *Amdo.101-Serpentes* might represent an intermediate lineage between the *Amdo-* and *Protoparvovirus* genera. However, this EPV exhibits several characteristic amdoparvoviral features including a putative M-ORF and a capsid gene that lacks a PLA2 domain [25]. Furthermore, *Amdo.101-Serpentes* groups more closely with amdo- than protoparvoviruses in the Rep phylogenies reconstructed here, supporting the view that it represents a reptilian lineage within an expanded *Amdoparvovirus* genus (**Fig 4**).

**Table 2. Incorporation of parvovirus DNA into the vertebrate germline.**

| *Parvovirus* genus | Host species group | | | | | | | | | |
|---|---|---|---|---|---|---|---|---|---|---|
| | Chondrichthyes species = 5 | | Actinopterygii species = 175 | | Sauria species = 200 | | Mammalia species = 353 | | Vertebrata species = 752* | |
| | loci | ratio | loci | ratio | loci | ratio | loci | ratio | loci | ratio |
| *Ichthamaparvovirus* | 0 | - | 1 | 0.01 | 2 | 0.01 | 0 | - | 2 | 0.003 |
| *Erythroparvovirus* | 0 | - | 0 | - | 0 | - | 2 | 0.01 | 2 | 0.003 |
| *Amdoparvovirus* | 0 | - | 0 | - | 1 | 0.01 | 2 | 0.01 | 3 | 0.004 |
| *Dependoparvovirus* | 0 | - | 0 | - | 13 | 0.07 | 66 | 0.19 | 80 | 0.108 |
| *Protoparvovirus* | 0 | - | 0 | - | 0 | - | 105 | 0.3 | 105 | 0.142 |
| Novel clades | 1 | *0.2* | 2 | 0.01 | 0 | - | 3 | 0.01 | 7 | 0.009 |
| Totals | 1 | *0.2* | 3 | 0.02 | 16 | 0.08 | 178 | 0.5 | 199 | 0.263 |

*Agnathans (*n* = 3), Amphibians (*n* = 15), and Lungfish (*Latimeria chalumnae*) were also screened, but results are not shown since no species in these groups were found to have EPVs. Ratio = unique loci/genomes screened.

**Table 3. Dates and age estimates used to calibrate parvovirus evolution.**

| *Parvovirus* lineage | Host species lineage(s) | High | Low |
|---|---|---|---|
| **Ortholog-based**[*] | | | |
| Primate AAVs (Dependo-A) | OW Primates | 23 | 16 |
| Neodependo- | Glires | 88 | 76 |
| Neodependo- | Lagomorpha | 77 | 23 |
| Neodependo- | Vespertilionidae | 49 | 38 |
| Neodependo- | Elephantidae | 23 | 9 |
| Neodependo- | Eulemur | 9 | 6 |
| Neodependo- | Hyracoidea | 14 | 7 |
| Lemuriadependo- | Whippomorpha | 56 | 52 |
| Lemuriadependo- | Rhinocerotidae | 51 | 15 |
| Lemuriadependo- | Phyllostomidae | 39 | 35 |
| Oceaniadependo- | Macropus | 45 | 27 |
| Amdo- | Serpentes | 111 | 100 |
| Amdo- | Hyracoidea | 14 | 7 |
| EDTC | Laurasitheria | 84 | 73 |
| Ichthama | Serpentes | 74 | 49 |
| Archaeo-Proto- | Ctenomyidae-Octodontidae | 24 | 16 |
| **Codivergence based**[**] | | | |
| Proto-Amdo lineage | Sharks/Bony fish | 497 | 450 |
| Neoprotoparvovirus | Eutherian mammals | 200 | 150 |
| Boca-Ave lineage | Aves/Mammalia | 393 | 297 |
| *Amdoparvovirus* | Sauria/Mammalia | 326 | 297 |
| *Protoparvovirus* | Actinopterygii/Mammalia | 446 | 425 |
| *Copiparvovirus* | Eutherian mammals | 111 | 100 |
| *Tetraparvovirus* | Eutherian mammals | 111 | 100 |
| *Erythroparvovirus* | Eutherian mammals | 111 | 100 |
| *Dependoparvovirus* | Euteleostii | 446 | 425 |
| **Biogeography-linked** | | | |
| *Protoparvovirus* | Mammalia/Pangaea | 200 | 180 |
| Archeoproto- NW rodent clade | NW Rodents/S. America | 50 | 30 |
| Erythroparvo- Rodent clade | Malagasy rodents/Madagascar | 30 | 20 |
| **U94 gene transfer** | | | |
| EDTC lineage | *Betaherpesvirinae* origins | 200 | 80 |

[*]Not all ortholog-based calibrations are shown, only the oldest for each virus lineage in which we identified orthologous sets of EPV sequences.

[**]Divergence dates obtained from TimeTree [35]. Complete records of ortholog-based dates can be found in **S5** and **S6** **Tables**.

Phylogenetic analysis of protoparvoviruses revealed previously unappreciated diversity within the *Protoparvovirus* genus: Three major subclades are present, which we labelled "Archaeoproto," "Mesoproto," and "Neoproto" (**Fig 6**). The "Archaeoproto" clade is comprised exclusively of EPVs and is highly represented in the genomes of Australian marsupials (Australidelphia), American marsupials (Ameridelphia), and New World rodents. The "Mesoproto" clade is also comprised exclusively of EPVs and was sparsely represented in the EPV fossil record, only being detected in the genomes of basal placental mammal groups (Xenarthra and Afrotheria). Finally, the "Neoproto" clade contains all known contemporary protoparvoviruses and a small number of EPV elements derived from these viruses (**Fig 6**).

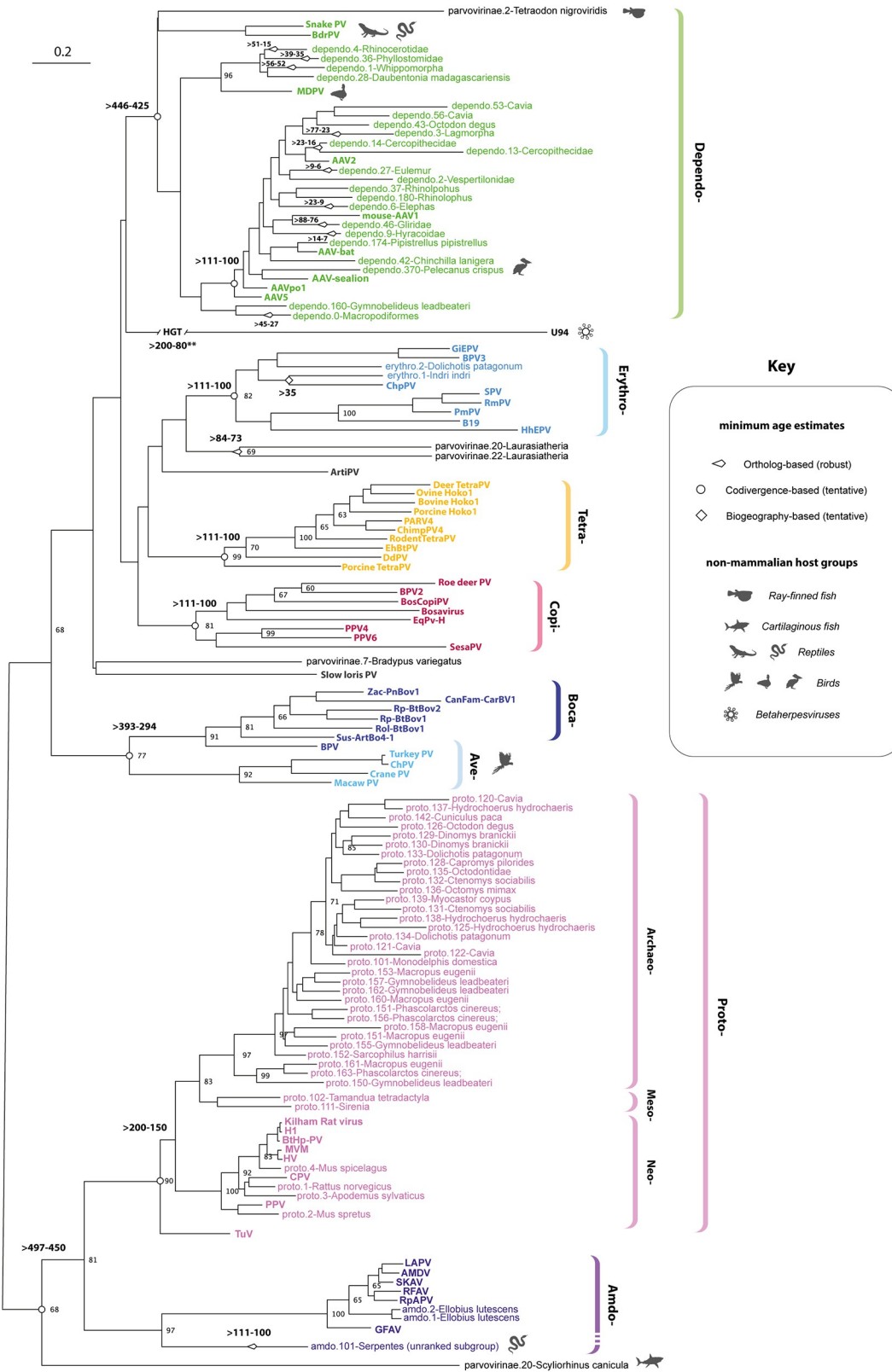

**Fig 4. Evolution of subfamily *Parvoviridae*.** An ML phylogeny showing the reconstructed evolutionary relationships between contemporary parvoviruses of subfamily *Parvovirinae* and the EPVs derived from subfamily *Parvovirinae*. The phylogeny, which is midpoint rooted for display purposes, was reconstructed using an MSA spanning 270 amino acid

residues in the Parvovirus Rep protein and the LG likelihood substitution model. Coloured brackets indicate the established parvovirus genera recognised by the International Committee for the Taxonomy of Viruses. Bootstrap support values (1,000 replicates) are shown for deeper internal nodes only. Scale bars show evolutionary distance in substitutions per site. Taxa labels are coloured based on taxonomic grouping as indicated by brackets; unclassified taxa are shown in black. Viral taxa are shown in bold, while EPV taxa are show in regular text. Numbers adjacent node shapes show minimum age estimates associated with lineages in millions of years before present (see **Table 3**). **Abbreviations**: AAV, adeno-associated virus; AMDV, Aleutian mink disease; BPV, bovine parvovirus; BrdPV, bearded dragon parvovirus; CPV, canine parvovirus; EPV, endogenous parvoviral element; HGT, horizontal gene transfer; HHV, human herpesvirus; MdPV, Muscovy duck parvovirus; ML, maximum likelihood; MSA, multiple sequence alignment; PV, Parvovirus. The data underlying this figure can be found at the following DOI: https://zenodo.org/record/6968218#.Yu115vHMIUY.

A novel, neoprotoparvovirus-derived EPV was identified in the steppe mouse (*Mus spicelagus*). Notably, the NS and VP genes of this EPV exhibit distinct phylogenetic relationships, implying recombination (S6F Fig). Furthermore, the VP/Cap gene of *proto.4-MusSpi* groups very closely with BtHp-PV, implying cross-species transmission (S8C and **S9** Figs). Rodent-associated taxa are interspersed throughout the "Neoproto" clade, and the neoprotoparvovirus-derived EPVs found in rodent genomes group with viruses isolated from carnivores, bats, and ungulates, rather than those isolated from rodents. Taken together, these phylogenetic relationships suggest that zoonotic transfer from rodents to other mammalian orders may occur relatively frequently among viruses in the "Neoproto" clade, as has been suggested for some retrovirus groups that infect mammals [29,30].

Phylogenetic reconstructions revealed the evolutionary relationships between dependo-related EPVs and contemporary dependoparvoviruses (**Figs 8A and S10**). The evolutionary origins of shorter and more degraded EPVs were more problematic to reconstruct. As might be expected, we obtained relatively low bootstrap support for internal branching relationships when such EPV sequences were included in the analysis (**S11 Fig**). However, if analysis is restricted to the longer EPVs, phylogenies disclose several robustly supported subclades within the *Dependoparvovirus* genus (**Fig 8A**). These included clades exclusive to reptilian species (Sauria-), Australian marsupials (Oceania-), and Boreoeutherian mammals (Neo-). A fourth clade, which we named "Shirdal," contains taxa derived from both avian and mammalian hosts.

Erythyroparvovirus-derived EPVs grouped with rodent erythroparvoviruses in phylogenetic trees, suggesting possible interorder transmission from rodents to lemuriforme primates (**S12 Fig**). When examined in relation to the biogeographic distribution of host species, these phylogenetic relationships provide tentative age calibrations for the *Erythroparvovirus* genus based on the parsimonious assumption that they spread to Madagascar and South America during the Cenozoic Era together with rodent founder populations (**Table 3**).

## Conservation of genome features in Parvovirinae evolution

We examined the distribution of conserved genome features among *Parvovirinae* genera in relation to the *Parvovirinae* phylogeny (**Fig 5**). For example, the "telomeres" that flank parvovirus genomes are heterotelomeric (asymmetrical) in some genera (*Amdo-*, *Proto-*, *Boca-*, and *Aveparvovirus*) whereas they are homotelomeric (symmetrical) in others [31]. Interestingly, the distribution of this trait across sublineages within the subfamily *Parvovirinae* suggests that the asymmetrical form (which is found across the "Amdo-Proto" and "Ave-Boca" sublineages) is more likely to be ancestral.

Similarly, in all *Parvovirinae* genera except *Aveparvovirus* and *Amdoparvovirus*, the N-terminal region of VP1 (the largest of the capsid) contains a phospholipase A2 (PLA2) enzymatic domain that becomes exposed at the particle surface during cell entry and is required for escape from the endosomal compartments. Phylogenetic reconstructions indicate that this

domain was present ancestrally and has been convergently lost in the *Aveparvovirus* and *Amdoparvovirus* genera (**Fig 5**) [2,32].

*Parvovirinae* genera also show variation in their gene expression strategies through differential promoter usage and alternative splicing. Members of the *Proto-* and *Dependoparvovirus* genera use two to three separate transcriptional promoters, whereas the *Amdo-*, *Erythro-*, and *Boca-* genera express all genes from a single promoter and use genus-specific read-through mechanisms to produce alternative transcripts [2,11]. Interestingly, both the *Proto-* and *Dependoparvovirus* genera utilise the first of these expression strategies despite being relatively distantly related, suggesting that the use of separate promoters could be the ancestral strategy within the subfamily *Parvovirinae*. However, this would mean that mechanisms to express multiple genes from a single promoter were acquired independently by the parvovirus genera that utilise them (**Fig 5**).

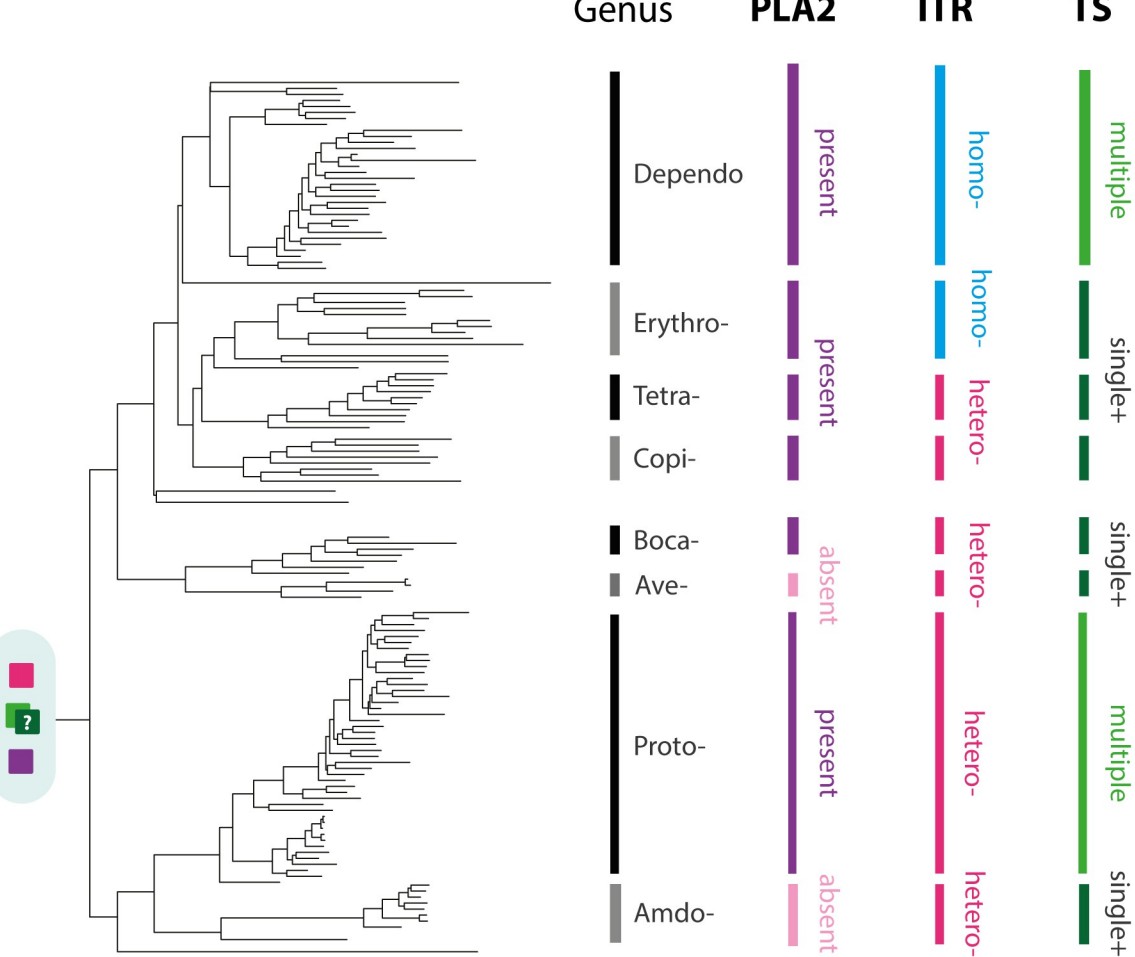

**Fig 5. Conservation of genome features during *Parvovirinae* evolution.** A midpoint rooted, ML phylogeny showing the reconstructed evolutionary relationships between contemporary parvoviruses of subfamily *Parvovirinae* and the ancient parvovirus species represented by EPVs. The phylogeny shown here is shown in greater detail in **Fig 4**. The black and grey vertical bars to the right of the phylogeny indicate parvovirus genera. Coloured bars indicate the distribution of virus traits across genera, following the key. The most likely ancestral state is indicated at the root of the tree, based on the parsimonious assumption that independent losses of genome features are more likely than independent gains. The ancestral TS remains unclear. **Abbreviations**: EPV, endogenous parvoviral element; Hetero, heterotelomeric; Homo, homotelomeric; ML, maximum likelihood; MTSP, multiple transcriptional start positions; STSP+, single transcription start position, plus additional strategies; TS, transcription strategy. The data underlying this figure can be found at the following DOI: https://zenodo.org/record/6968218#.Yu115vHMIUY.

## Mammalian vicariance has shaped the evolution of protoparvoviruses

The recovery of a rich fossil record for protoparvoviruses allowed us to examine how their evolution has been shaped by macroevolutionary processes impacting on mammals over the past 150 to 200 My, such as continental drift [33]. Around 200 Mya, the supercontinent of Pangaea, then the sole landmass on the planet, began separating into two subcomponents (Fig 7). One (Laurasia) comprised Europe, North America, and most of Asia, while the second (Gondwanaland) comprised Africa, South America, Australia, India, and Madagascar. Mammalian subpopulations were fragmented by these events, and then fragmented further as Gondwanaland separated into its component continents. The associated genetic isolation due to geographic separation (vicariance) drove the early diversification of major subgroups, including indigenous mammalian lineages in South America (xenarthans and marsupials), Australia (marsupials), and Africa (afrotherians). At points throughout the Cenozoic Era, placental mammal groups that evolved in Laurasia (boreoeutherians) expanded into other continental regions. For example, the ancestors of contemporary New World rodents (which include capybaras, chinchillas, and guinea pigs among many other, highly diversified species) are thought to have reached the South American continent approximately 35 Mya [34].

Protoparvoviruses phylogenies strikingly reflect the impact of mammalian vicariance—and later migration—on protoparvovirus emergence and spread during the Cenozoic Era. When protoparvovirus-related EPVs are included in ML-based reconstructions, the internal structure of the resultant phylogeny has extremely robust support (Fig 6). Moreover, this phylogeny can readily be mapped onto a phylogeny of mammals (obtained via TimeTree; [35]) so that the three major protoparvovirus lineages emerge in concert with major groups of mammalian hosts (Fig 7C). Importantly, however, one exception to this pattern occurs in the "Archeoproto" clade in which EPVs from New World rodent genomes group with EPVs found in marsupial genomes, with the closest relatives being EPVs identified in the common opossum (*Monodelphis domestica*), a South American marsupial (Fig 6). We propose that, as shown in Fig 7, these relationships can be accounted for by a parsimonious model of protoparvovirus evolution wherein (i) ancestral protoparvovirus species were present in Pangaea prior to its breakup; (ii) vicariance among ancestral mammal populations led to the emergence of distinct protoparvovirus clades in distinct biogeographic regions, with the "archeoprotoparvovirus" (ArcPV) clade evolving in marsupials, and the "meso-" and "neo-" clades evolving in placental mammals; and (iii) founding populations of New World rodents were exposed to infection with ArcPVs following rodent colonisation of the South American continent (estimated to have occurred approximately 50 to 30 Mya; [34]). This simple model can account for the phylogenetic relationships shown in Fig 6, as well as the high frequency of ArcPV-derived EPVs in the genomes of New World rodent species versus their complete absence from the genomes of Old World rodent species.

## Interclass transmission and the evolution of dependoparvoviruses

Phylogenies imply a role for interclass transmission between mammals and birds in dependoparvovirus evolution (Fig 8). Firstly, in both midpoint-rooted phylogenies, and in phylogenies rooted on the saurian dependoparvoviruses (as proposed by Penzes and colleagues; [36]), the "Shirdal" clade falls intermediate between two exclusively mammalian groups—the nonautonomous AAVs found in placental mammals, and clade Oceania—found exclusively in Australian marsupials (Fig 8A). This implies an ancestral switch from mammalian to avian hosts (green arrow; Fig 8B). Furthermore, the avian viruses in this clade group basally (Ave-), forming a paraphyletic group relative to a derived subclade ("Lemuria") of EPVs obtained from a diverse range of mammalian hosts. This implies a second, subsequent jump from birds to mammals (blue arrow; Fig 8B).

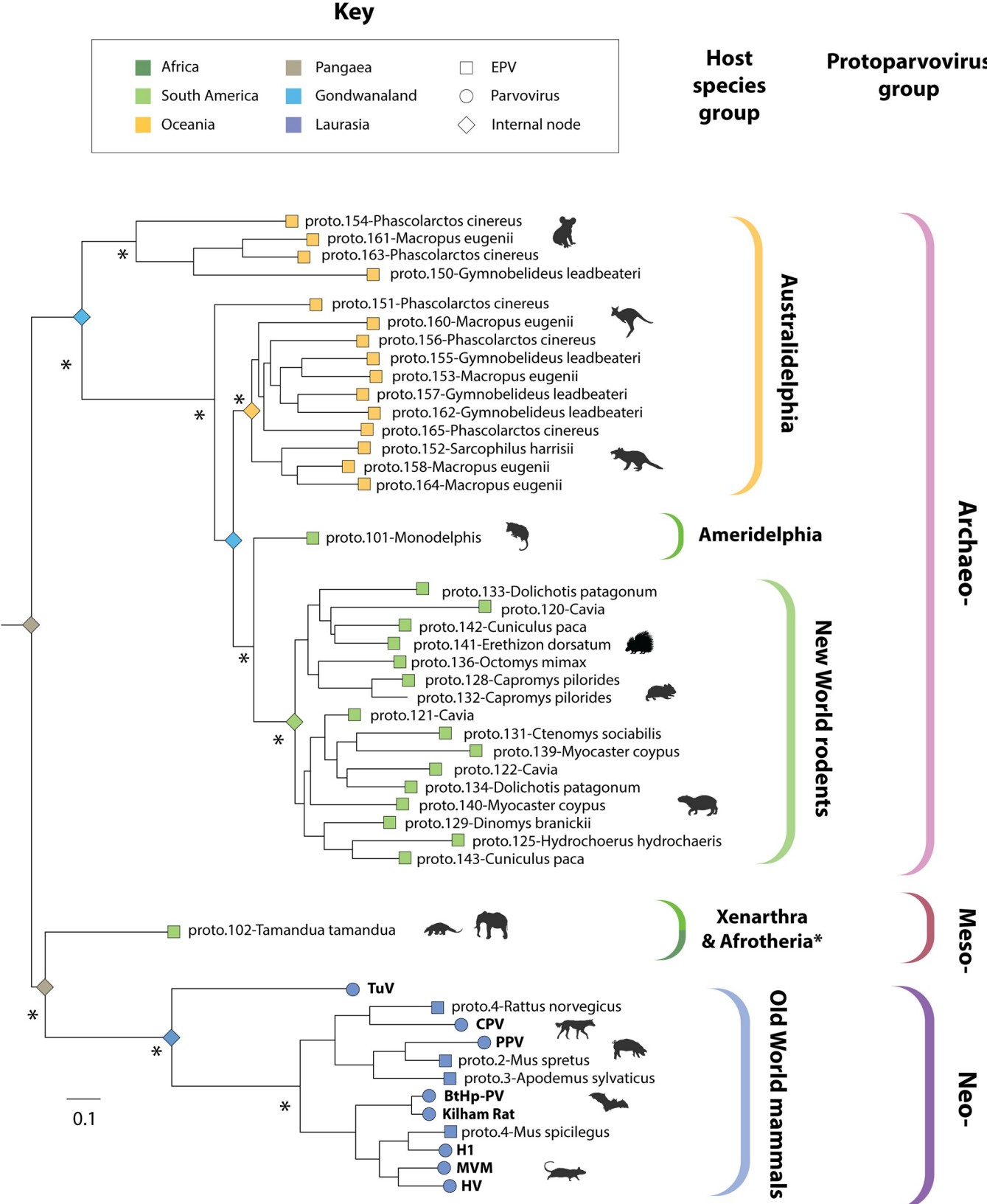

**Fig 6. Phylogenetic relationships of protoparvoviruses and protoparvovirus-like EPVs.** An ML-based phylogeny showing the reconstructed evolutionary relationships between contemporary protoparvovirus species and the ancestral protoparvovirus species represented by EPVs. The phylogeny was constructed from an MSA spanning 712 amino acid residues in the Rep protein (substitution model = LG likelihood) and is midpoint rooted for display purposes. Asterisks indicate nodes with bootstrap support >85% (1,000 replicates). The scale bar shows evolutionary distance in substitutions per site. Coloured brackets to the right indicate the following: (i) robustly supported subclades within the *Protoparvovirus* genus (outer set of brackets) and (ii) the implied host range of each subclade (inner set of brackets). Terminal nodes are represented by squares (EPVs) and circles (viruses) and are coloured based on the biogeographic distribution of the host species in which they were identified (see key). Coloured diamonds on internal nodes show the biogeographic distribution of host species ancestors (based on fossil evidence) [33]. *Phylogenetic evidence for the presence of "mesoprotoparvoviruses" in Afrotherian species is presented in **Fig 4**. EPV, endogenous parvoviral element; ML, maximum likelihood; MSA, multiple sequence alignment.

The "Neodependo" clade is comprised of AAVs and EPVs related to AAVs, all of which occur exclusively in placental mammals. AAVs are sometimes referred to as "nonautonomous dependoparvoviruses" because they require the presence of a "helper" virus to replicate [1]. In Rep phylogenies, the "Neodependo" clade groups in a derived position relative to clades containing the autonomously replicating dependoparvoviruses of birds and reptiles (**Figs 4 and 8**). These observations indicate that dependency is an ancestral, shared characteristic of AAVs and is likely to have evolved in placental mammal hosts.

## Coding capacity and expression of EPV sequences

Previous studies have shown that some EPV loci express RNA with the potential to encode polypeptide gene products, either as unspliced viral RNA [17,37,38] or as fusion genes comprising RNA sequences derived from both host and viral sources [39]. We examined all EPV loci identified in our study to determine their coding potential. We identified 56 unique loci at which sequences encoding uninterrupted polypeptide sequences of 300 amino acids (aa) or more are retained (**Table 4**). Extended regions of intact coding sequence have been maintained in a diverse collection of EPVs, including some that are demonstrably many millions of years old (e.g., *amdo.101-Serpentes*). mRNA expression has been experimentally demonstrated for several EPVs [18,37,38], and screening of RNA databases revealed evidence for expression of RNA from several additional EPV loci (**Table 4**).

## Discussion

In this study, we used EPVs to investigate the long-term coevolutionary interactions between parvoviruses and vertebrates. We recovered the complete repertoire of EPV sequences in WGS data representing 752 vertebrate species. While previous studies have reported a sampling of EPV diversity in vertebrates [12–18,25,37,39–42], our investigation is an order of magnitude larger in scale: We identify 364 sequences representing nearly 200 discrete germline incorporation events that took place during the Cenozoic Era (**Fig 2 and S4–S6 Tables**). Furthermore, we introduced a higher level of order to these data by (i) discriminating between unique loci and orthologous copies; (ii) aligning EPVs to contemporary viruses and hierarchically arranging MSAs so that phylogenetic and genomic comparisons could utilise the maximum amount of available data; and (iii) applying a standardised nomenclature to EPVs that captures information about orthology and taxonomy.

Our analysis shows that parvovirus DNA was incorporated into the vertebrate germline throughout the Cenozoic Era (**Fig 3**). The independent formation and fixation of EPVs in such a diverse range of taxa shows that multiple *Parvovirinae* genera circulated widely among vertebrate fauna during their evolution. Furthermore, the robust calibrations we obtain from EPVs lend credibility to more tentative, biogeography and distribution-based age estimates obtained for other parvovirus lineages that are not represented in the genomic "fossil record" (**Fig 4** and **Table 3**). Given that the origins of the parvovirus family likely extend far back into

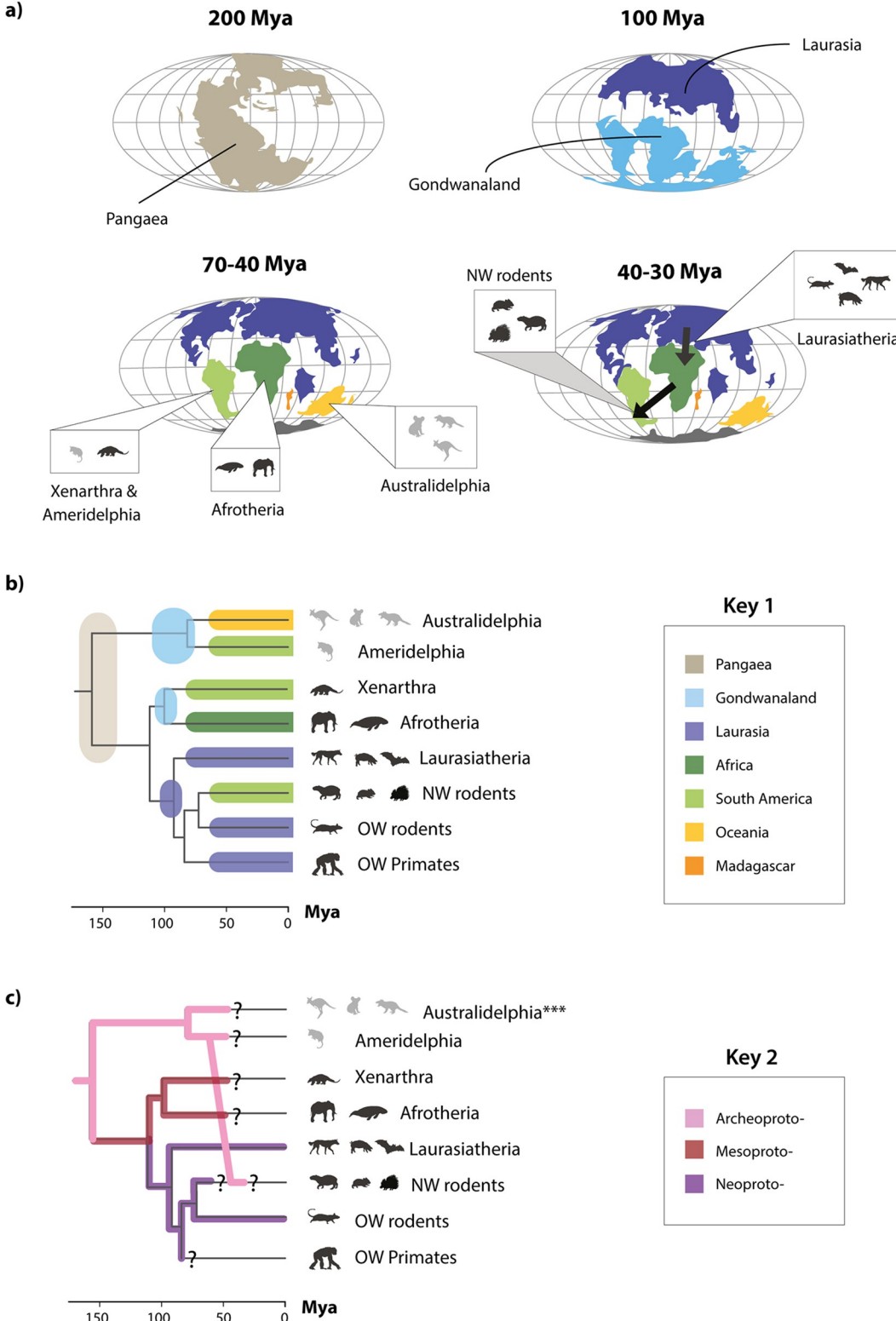

**Fig 7. Protoparvovirus evolution has been shaped by mammalian vicariance. (a)** Mollweide projection maps showing how patterns of continental drift from 200–35 led to periods of biogeographic isolation for terrestrial mammals in Laurasia (Europe and Asia), South America, Australia, Africa, and Madagascar. The resulting vicariance is thought to have contributed to the diversification of mammals, reflected in the mammalian phylogeny as shown in panel (b). Most placental

mammals (including rodents, primates, ungulates, and bats) evolved in Laurasia. However, these groups later expanded into other continents, and fossil evidence indicates that the ancestors of today's "New World rodents" had arrived on the South American continent by approximately 35 Mya, if not earlier. Plate tectonic maps were downloaded from ODSN Plate Tectonic Reconstruction Service (https://www.odsn.de/odsn/services/paleomap/paleomap.html). **(b)** A time-calibrated phylogeny of mammals (obtained via TimeTree; [35]) with annotations indicating the biogeographic associations of the major taxonomic groups of contemporary mammals and ancestral mammalian groups, following panel (b) and key 1. **(c)** A time-calibrated phylogeny of mammals (obtained via TimeTree; [35]) annotated to indicate the inferred distribution of protoparvovirus subgroups among mammalian groups, following key 2. Question marks indicate where it is unknown if viral counterparts of the lineages represented by EPVs still circulate among contemporary members of the host species groups in which they are found. The data underlying this figure can be found in https://zenodo.org/record/6968218#.Yu115vHMIUY. **Abbreviations**: CPV, carnivore parvovirus type 1; HV, hamster parvovirus; Mya, millions of years ago; NW, New World; ODSN, Ocean Drilling Stratigraphic Network; OW, Old World; PPV, porcine parvovirus; TuV, Tusavirus.

the early evolutionary history of animal species [16,43], it is reasonable to propose that sub-family *Parvovirinae* could have emerged in broad congruence with the diversification of major vertebrate groups. Indeed, this extended evolutionary timeline can be strikingly visualised in the *Protoparvovirus* genus, in which the emergence and spread of sublineages reflects the impact of vicariance and continental drift on mammal evolution during the Cenozoic Era (**Figs 6 and 7**). A limitation of our study is that it relied on opportunistic sampling via published whole genome data—we expect that more representative sampling of vertebrate species genomes will reveal new information, particularly regarding more recently integrated ERVs.

The extent to which vertebrate EPVs have reached fixation through positive selection as opposed to incidental factors such as founder effects, population bottlenecks, and genetic hitchhiking remains unclear. Potentially, EPVs might sometimes be co-opted or "exapted" as has been reported for EVEs derived from other virus groups [44–46]. Recent studies have identified EPVs in the germline of degus (*Octodon degus*) and elephants (family Elephantidae) that encode intact Rep ORFs and exhibit similar patterns of tissue-specific mRNA expression in the liver [37,38], suggesting that expression of Rep protein or mRNA may be physiologically relevant in mammals. We find that the coding capacity of the VP/capsid ORF is also strikingly conserved in some EPVs (**Table 4**). Potentially, EPVs capable of encoding protein products could function as antiviral immune factors capable of blocking infection with related viruses [44].

Notably, multiple distinct genera of parvoviruses are often found infecting the same species groups—for example, at least seven distinct genera circulate in mammals. Furthermore, EPV-based calibrations indicate that these genera are likely to have cocirculated among mammals for many millions of years (**Fig 3**). The extended evolutionary timeline implied our analysis is consistent with the idea that the persistence of multiple, distinct parvovirus genera in the same host species groups reflects adaptive divergence among these genera, such that each parvovirus occupies a distinct "ecological niche" (i.e., part of the ecological space available in the environment) [47]. Although niches can be difficult to define precisely, most treatments consider conditions of the physical environment, characteristics of resources, and the traits of other interacting species as important factors [48]. For viruses, host species range is inevitably a major influence, but other aspects of replication could also come into play. While all members of the subfamily *Parvovirinae* use similar basic mechanisms to achieve specific steps in infection, the details of these processes (e.g., the specific cell types and organ systems targeted for replication) frequently differ between genera. For example, primate erythroparvoviruses target erythroid progenitor cells [3], mammalian chaphamaparvoviruses are suspected to be nephrotropic [49], and antibody-dependent enhancement is suspected to be a shared characteristic of amdoparvoviruses [50,51].

The nonautonomous parvoviruses or "AAVs" provide an interesting example of adaptation to a specialised niche. Superficially, the requirement for a helper virus appears to be a

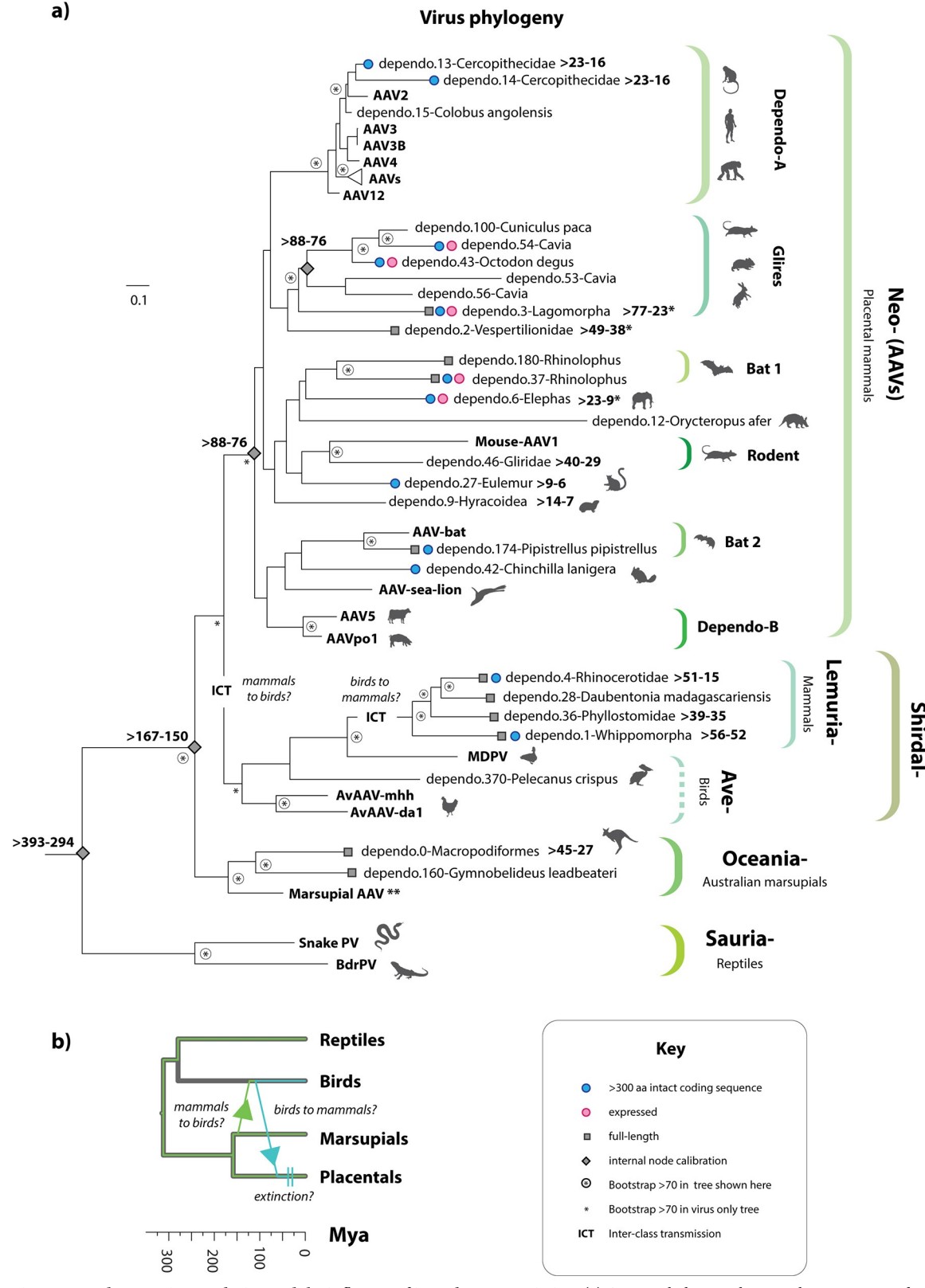

**Fig 8. Dependoparvovirus evolution and the influence of interclass transmission. (a)** An ML phylogeny showing the reconstructed evolutionary relationships between contemporary dependoparvovirus species and the ancient dependoparvovirus species represented by

EPVs. Virus taxa names are shown in bold; EPVs are shown in regular text. The phylogeny was constructed from an MSA spanning 330 amino acid residues of the Rep protein and the LG likelihood substitution model and is rooted on the reptilian lineage. Brackets to the right indicate proposed taxonomic groupings. Shapes on leaf nodes indicate full-length EPVs and EPVs containing intact/expressed genes (see key). Numbers next to leaf nodes indicate minimum age calibrations for EPV orthologs. Shapes on branches and internal nodes indicate different kinds of minimum age estimates for parvovirus lineages, as shown in the key. Numbers adjacent node shapes show minimum age estimates associated with lineages in millions of years before present (see **Table 3**). For taxa that are not associated with mammals, organism silhouettes indicate species associations, as shown in the key. The scale bar (top left) shows evolutionary distance in substitutions per site. Asterisks in circles indicate nodes with bootstrap support >70% (1,000 replicates). Plain asterisks indicate nodes that are not supported in the tree shown here but are supported in phylogenies based on longer regions of Rep (**S7 Fig**). *Age calibrations based on data obtained in references [18,38]. **A contemporary virus derived from the marsupial clade has been reported in marsupials, but only transcriptome-based evidence is available [17]. **(b)** A time-calibrated phylogeny of vertebrate lineages showing proposed patterns of interclass transmission within the "Shirdal" clade. **Abbreviations**: aa, amino acid residues; AAV, adeno-associated virus; BrdPV, bearded dragon parvovirus; EPV, endogenous parvoviral element; MdPV, Muscovy duck parvovirus; ML, maximum likelihood; MSA, multiple sequence alignment; ORF, open reading frame; PV, Parvovirus. The data underlying all panels in this figure can be found in https://zenodo.org/record/6968218#.Yu115vHMIUY.

deficiency rather than adaptation. However, molecular genetic analysis of AAV2 indicates that helper virus–dependent replication resulted from a gain of function rather than loss of autonomous replication [52,53]. Conceivably, dependency could have evolved as a means of optimising the probability of successful transmission. The "helper" virus groups that enable AAV replication (e.g., adenoviruses, herpesviruses) have large double-stranded DNA genomes and can therefore encode more complex means of sensing environmental conditions in their genomes, only replicating when conditions are optimal [54]. By "tethering" their replication cycle to that of these larger, more sophisticated viruses, AAVs can perhaps take advantage of their capacities.

Our investigation shows that vertebrate parvoviruses have ancient associations with their hosts and have acquired lineage-specific adaptations over many millions of years. Importantly, this implies that the growing wealth of genomic data—incorporating both virus and EPV sequences—can be harnessed to define the biological characteristics of parvovirus species and groups. By linking our growing knowledge of parvovirus distribution, diversity, and evolution with experimental and field studies, we can now develop parvovirus-based therapeutics that are grounded in an understanding of natural history. Such comparative, evolutionary approaches will not only help identify virus species with desirable characteristics—e.g., a well-defined tropism for a specific cell type—they establish a powerful, rational basis for dissecting structure–function relationships in parvovirus genomes. To support the broader use of evolution-related domain knowledge in parvovirus research, we published our data in the form of an open, extensible database framework (**S1 Fig**). We hope that by enabling the reproduction of comparative genomic analyses and supporting reuse of the complex datasets that underpin them, these resources can help researchers exploit genomic data to develop a deeper understanding of parvovirus biology.

## Methods

### Screening in silico of whole genome sequence databases

We used the Database-Integrated Genome Screening (DIGS) tool [55] to derive a nonredundant database of EPV loci within published WGS assemblies. The DIGS tool is a Perl-based framework in which the Basic Local Alignment Search Tool (BLAST) program suite [56] is used to perform similarity searches and the MySQL relational database management system to coordinate screening and record output data. A user-defined reference sequence library provides (i) a source of "probes" for searching WGS data using the tBLASTn program; and (ii) a means of classifying DNA sequences recovered via screening (**S5 Fig**). For the purposes of the

**Table 4. Coding capacity and expression of vertebrate EPVs.**

| Genus | Clade[a] | Number[b] | Organism[c] | Exp.[d] | Length (aa)[e] | Genes[f] |
|---|---|---|---|---|---|---|
| Amdo- | | 1 | *Ellobius lutescens* | *p* | 383 | NS |
| Amdo- | | 2 | " | | **586** | NS-MAFG* |
| Amdo- | | 101 | *Protobothrops mucrosquamatus* | *p* | **574** | NS |
| Amdo- | | 101 | " | | 333 | **VP** |
| Dependo- | Oceania- | 0 | *Macropus eugenii* | | 442 | **VP** |
| Dependo- | Lemuria- | 1 | *Orcinus orca* | | 357 | **VP** |
| Dependo- | Lemuria- | 1 | *Globicephala melas* | | 314 | NS |
| Dependo- | Neo- (AAV) | 2 | *Myotis lucifugus* | *p* | 311 | NS |
| Dependo- | Neo- (AAV) | 3 | *Lepus timidus* | *c* | 439 | NS |
| Dependo- | Neo- (AAV) | 4 | *Dicerorhinus sumatrensis* | | 321 | **VP** |
| Dependo- | Neo- (AAV) | 6 | *Loxodonta africana* | *c* | **583** | NS |
| Dependo- | Neo- (AAV) | 13 | *Cercocebus atys* | | 309 | NS |
| Dependo- | Neo- (AAV) | 14 | " | | **561** | NS |
| Dependo- | Neo- (AAV) | 27 | *Eulemur macaco* | | 473 | NS |
| Dependo- | Neo- (AAV) | 37 | *Rhinolophus sinicus* | | 382 | NS |
| Dependo- | Neo- (AAV) | 42 | *Chinchilla lanigera* | | **499** | NS |
| Dependo- | Neo- (AAV) | 43 | *Octodon degus* | *c* | **502** | NS |
| Dependo- | Neo- (AAV) | 54 | *Cavia porcellus* | *c* | <u>230</u> | NS-Myo9* |
| Dependo- | Neo- (AAV) | 59 | *Myocastor coypus* | | 304 | NS |
| Dependo- | Neo- (AAV) | 88 | *Megaderma lyra* | | 330 | NS |
| Dependo- | Neo- (AAV) | 174 | *Pipistrellus pipistrellus* | | 449 | NS |
| Dependo- | Neo- (AAV) | 174 | " | | **634** | **VP** |
| Dependo- | Sauria- | 201 | *Thamnophis elegans* | | **722** | **VP** |
| Dependo- | Sauria- | 201 | " | | 385 | **VP** |
| Dependo- | Sauria- | 202 | " | | 404 | **VP** |
| Erythro- | | 1 | *Indri indri* | | **701** | NS |
| Erythro | | 1 | " | | **567** | **VP** |
| Ichthama- | | 2 | *Emydocephalus ijimae* | | 478 | ORF2 |
| Ichthama- | | 2 | *Hydrophis melanocephalus* | | 392 | NS |
| Proto- | Neo- | 1 | *Rattus norvegicus* | | 440 | **VP** |
| Proto- | Neo- | 2 | *Mus spretus* | | 301 | NS |
| Proto- | Neo- | 2 | " | | 484 | **VP** |
| Proto- | Neo- | 3 | *Apodemus sylvaticus* | | 341 | **VP** |
| Proto- | Neo- | 3 | " | | 314 | NS |
| Proto- | Neo- | 4 | *Mus spicilegus* | | 403 | NS |
| Proto- | Neo- | 4 | " | | **714** | **VP** |
| Proto- | Meso- | 102 | *Tamandua tetradactyla* | | **563** | NS |
| Proto- | Meso- | 102 | " | | **686** | **VP** |
| Proto- | Archaeo- | 103 | *Monodelphis domestica* | | 349 | **VP** |
| Proto- | Archaeo- | 107 | *Myocastor coypus* | | 486 | NS |
| Proto- | Archaeo- | 107 | " | | **589** | **VP** |
| Proto- | Archaeo- | 108 | *Hydrochoerus hydrochaeris* | | 371 | NS |
| Proto- | Archaeo- | 131 | *Ctenomys sociabilis* | | 346 | NS |
| Proto- | Archaeo- | 131 | " | | 473 | **VP** |
| Proto- | Archaeo- | 134 | *Dolichotis patagonum* | | 337 | NS/VP |
| Proto- | Archaeo- | 134 | " | | 387 | **VP** |
| Proto- | Archaeo- | 138 | *Hydrochoerus hydrochaeris* | | 429 | **VP** |

*(Continued)*

**Table 4.** (Continued)

| Genus | Clade[a] | Number[b] | Organism[c] | Exp.[d] | Length (aa)[e] | Genes[f] |
|-------|----------|-----------|-------------|---------|----------------|----------|
| Proto- | Archaeo- | 139 | *Myocastor coypus* | | 334 | **VP** |
| Proto- | Archaeo- | 141 | *Erethizon dorsatum* | | 361 | NS |
| Proto- | Archaeo- | 143 | *Cuniculus paca* | | 462 | NS/VP |
| Proto- | Archaeo- | 152 | *Sarcophilus harrisii* | | 363 | NS |
| Proto- | Archaeo- | 155 | *Gymnobelideus leadbeateri* | | 320 | **VP** |
| Proto- | Archaeo- | 158 | *Macropus eugenii* | | 429 | **VP** |
| Proto- | Archaeo- | 178 | *Erethizon dorsatum* | | 308 | NS |
| Proto- | Archaeo- | 189 | *Grammomys surdaster* | *p* | <u>82</u> | NS |
| Proto- | Archaeo- | 210 | *Macropus eugenii* | | 327 | **VP** |
| Proto- | Archaeo- | 210 | " | | 314 | NS |

We measured unbroken coding regions within the range of established open reading frames and did not require a methionine start codon for any of these coding "sections."

[a]Subclades are as shown in **Figs 5** and 6.

[b]Locus numeric ID.

[c]Latin binomial of species in which ortholog containing longest coding sequence was detected.

[d]*p* = computationally predicted, *c* = confirmed via polymerase chain reaction. aa = amino acid residues.

[e]Sequences are shown that encode $>= 300$ amino acids of coding sequence or have evidence of expression. Sequences that are <300 aa in length but with evidence for expression are underlined.

[f]VP shown in bold.

*EPV–gene fusion products predicted [25] or confirmed [39]—length shown is for viral portion only.

present project, we collated a reference library composed of polypeptide sequences derived from representative parvovirus species and previously characterised EPVs. WGS data of animal species were obtained from the National Center for Biotechnology Information (NCBI) genome database [57]. We obtained all animal genomes available as of March 2020. We extended the core schema of the screening database to incorporate additional tables representing the taxonomic classifications of viruses, EPVs, and host species included in our study. This allowed us to interrogate the database by filtering sequences based on properties such as similarity to reference sequences, taxonomy of the closest related reference sequence, and taxonomic distribution of related sequences across hosts. Using this approach, we categorised sequences into the following: (i) putatively novel EPV elements; (ii) orthologs of previously characterised EPVs (e.g., copies containing large indels); and (iii) nonviral sequences that cross-matched to parvovirus probes. Sequences that did not match to previously reported EPVs were further investigated by incorporating them into genus-level, genome-length MSAs (see **Table 1**) with representative parvovirus genomes and reconstructing ML phylogenies using RAxML (version 8.2.12) [58]. Where phylogenetic analysis supported the existence of a novel EPV insertion, we also attempted to (i) determine its genomic location relative to annotated genes in reference genomes; and (ii) identify and align EPV–host genome junctions and preintegration insertion sites. Where these investigations revealed new information (e.g., by confirming the presence of a previously uncharacterised EPV insertion), we updated our reference library accordingly. This in turn allowed us to reclassify EPV loci in our database and group sequences more accurately into categories. By iterating this procedure, we progressively resolved the majority of EPV sequences identified in our screen into groups of orthologous sequences derived from the same initial germline incorporation event (**S4–S6 Tables**).

## Creation of resources for reproducible comparative analysis of parvovirus genomes

We used GLUE, a bioinformatics software framework, to develop open data resources for parvoviruses [20]. A library of parvovirus reference sequences (**S1 Table**) incorporating all species recognised by the International Committee for Taxonomy of Viruses (ICTV) was obtained from GenBank. A standard set of genome features was defined based on published literature and GenBank annotations (**S2 Table**). We selected well-annotated, high-quality reference genomes as "master" reference sequences for each subfamily, genus, and species represented in our MSA hierarchy (**Table 1**) and annotated the locations of genome features within these reference genomes. MSAs were constructed using MUSCLE [59] and GLUE's native BLAST-based aligners [56,60].

Isolate data were captured by extending GLUE's underlying database schema. GenBank sequences in XML format are imported into the Parvovirus-GLUE project using GLUE's "genbankImporter" module to extract sequence and isolate-associated data. Nonstandard fields (e.g., isolate-specific information in the "notes" section of the GenBank entry) were extracted using a regular expression library, and a FreeMarker template was used to standardise their values as described previously [19]. Additional data (e.g., information missing from GenBank records but identified in an associated publication) were imported from tabular files using GLUE's TextFilePopulator module. Where (non-master) reference genome sequences were lacking feature annotations, we used GLUE's "inherit features" command [19] to infer their coordinates from an MSA in which the genomic coordinate space was constrained to a fully annotated master reference sequence for the corresponding genus (see **Table 1**).

GLUE allows MSAs to be hierarchically linked via a "constrained alignment tree" data structure [19] (**Table 1 and S3 Fig**). In each MSA, a chosen master reference sequence constrains the genomic coordinate space. For contemporary parvoviruses, we used our chosen genus master reference sequences, while for EPVs, we used consensus EPVs as constraining references for MSAs representing orthologous EPV loci. Each MSA in the Parvovirus-GLUE MSA hierarchy is linked to its child and/or parent MSAs via our chosen set of references (**Table 1**)—this creates in effect a single, unified MSA that can be used to implement comparisons across a range of taxonomic ranks while also making use of the maximum amount of available information at each level. By standardising the genomic coordinate space to the constraining reference sequence selected for each MSA, the alignment tree enables standardised genome comparisons across the entire *Parvoviridae* family. Note that, while MSAs representing internal nodes (see **Table 1**) contain only master reference sequences, they can be recursively populated with all taxa contained in child alignments when exported using any of GLUE's native exporter modules [19].

MSA partitions derived from the constrained MSA tree (**Table 1**) were used as input for phylogenetic reconstructions. We used feature coverage information, generated for each aligned sequence, to condition the selection of taxa into MSA partitions for phylogenetic analysis. The "record feature coverage" function of GLUE was used to generate coverage data for all members of all MSAs. Coverage data were generated for all genome features annotated in the constraining reference sequence.

To accommodate EPV data in Parvovirus-GLUE, we extended the underlying database schema to incorporate an EPV-specific table with data fields capturing EPV characteristics (e.g., locus coordinates, ortholog group, flanking genes). In addition, where multiple EPV orthologs were identified, we created MSAs to represent homology between individual orthologs in each EPV set, we used these to (i) reconstruct the evolutionary relationships between orthologs [20]; and (ii) derive consensus reference sequences for each EPV locus.

## Genomic analysis of EPVs and viruses

Putative ancestral ORFs of EPVs were inferred by manual comparison to parvovirus reference genomes. The putative peptide sequences of EVEs (i.e., the virtually translated sequences of EVE ORFs, repaired to remove frameshifting indels) were then aligned with the polypeptide sequences encoded by reference genomes, using MUSCLE. Protein-level phylogenies were reconstructed using ML as implemented in RAxML (version 8.2.12) [58]. Protein substitution models were selected via hierarchical ML ratio test using the PROTAUTOGAMMA option. For multicopy EPV lineages, we constructed MSAs and phylogenetic trees to confirm that branching relationships follow those of host species (S4B Fig; [20]). Phylogenies of EPV orthologs were reconstructed using ML as implemented in RAxML [58] and the GTR model of nucleotide selection as selected using the likelihood ratio test. Time-calibrated vertebrate phylogenies were obtained via TimeTree, an open database of species divergence time estimates [35].

## Expression and intactness of EPVs

We identified open coding regions of coding sequence in EPVs using scripts included with Parvovirus-GLUE [20]. To determine if there was evidence of expression of EPVs in host species, we searched the NCBI Reference RNA Sequences (refseq_rna) with Dependoparvovirus VP and Rep sequences (NC_002077). We used a translated nucleotide query and a translated database using tBLASTx [56] and evaluated alignments found between refseq_rna sequences and Dependoparvovirus VP and Rep sequences. To further verify expression, we determined if the annotations were solely based on computational prediction or RNAseq alignment annotations. For host species with evidence of expression, we conducted BLASTn searches within refseq_rna to identify expressed EPVs.

## Supporting information

**S1 Fig. Parvovirus-GLUE—An open resource for reproducible comparative analysis of parvovirus genome data.**
(DOCX)

**S2 Fig. The Parvovirus-GLUE resource build process.**
(DOCX)

**S3 Fig. A constrained alignment tree for the *Parvoviridae.***
(DOCX)

**S4 Fig. Genome screening in silico.**
(DOCX)

**S5 Fig. Phylogeny construction using Parvovirus-GLUE.** Flowcharts showing the process through which maximum likelihood phylogenies of endogenous parvoviral elements (EPVs) were constructed using Parvovirus-GLUE. The data underlying this figure can be found in https://zenodo.org/record/6968218.
(DOCX)

**S6 Fig. Phylogeny construction using Parvovirus-GLUE.**
(DOCX)

**S7 Fig. Comprehensive phylogenetic analysis of subfamily *Parvovirinae* using virus sequences only.**
(DOCX)

**S8 Fig. Comprehensive phylogenetic analysis of subfamily *Parvovirinae* including virus and EPV sequences.**
(DOCX)

**S9 Fig. Evolution of protoparvoviruses.**
(DOCX)

**S10 Fig. Dependoparvovirus VP/capsid phylogeny.**
(DOCX)

**S11 Fig. Multiple sequence alignment of dependoparvovirus-derived EPVs and dependo-parvoviruses.**
(DOCX)

**S12 Fig. Evolution of the *Erythroparvovirus* genus.**
(DOCX)

**S13 Fig. Evolutionary relationships of vertebrate hamaparvoviruses.**
(DOCX)

**S1 Table. Parvovirus reference genomes included in Parvovirus-GLUE.**
(DOCX)

**S2 Table. Parvovirus genome features defined in Parvovirus-GLUE.**
(DOCX)

**S3 Table. Locations of genome features within parvovirus reference sequences.**
(DOCX)

**S4 Table. Vertebrate endogenous parvoviral elements identified derived from dependopar-voviruses.**
(DOCX)

**S5 Table. Vertebrate endogenous parvoviral elements identified derived from protoparvo-viruses.**
(DOCX)

**S6 Table. Vertebrate endogenous parvoviral elements that are not derived from proto- or dependoparvoviruses.**
(DOCX)

## Author Contributions

**Conceptualization:** Robert M. Kotin, Robert J. Gifford.

**Data curation:** Matthew A. Campbell, Shannon Loncar, Robert J. Gifford.

**Formal analysis:** Matthew A. Campbell, Shannon Loncar, Robert J. Gifford.

**Funding acquisition:** Robert M. Kotin.

**Investigation:** Matthew A. Campbell, Shannon Loncar, Robert J. Gifford.

**Methodology:** Matthew A. Campbell, Shannon Loncar, Robert J. Gifford.

**Project administration:** Robert M. Kotin.

**Resources:** Robert J. Gifford.

**Software:** Matthew A. Campbell, Robert J. Gifford.

**Supervision:** Robert M. Kotin.

**Validation:** Shannon Loncar, Robert J. Gifford.

**Visualization:** Matthew A. Campbell, Shannon Loncar, Robert J. Gifford.

**Writing – original draft:** Robert J. Gifford.

**Writing – review & editing:** Matthew A. Campbell, Robert M. Kotin, Robert J. Gifford.

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
