## [Editor Report · Decision Letter 0]

4 May 2022

Dear Dr Gifford, 

Thank you for submitting your manuscript entitled "Comparative analysis reveals the long-term co-evolutionary history of parvoviruses and vertebrates." for consideration as a Research Article by PLOS Biology.

Your manuscript has now been evaluated by the PLOS Biology editorial staff, as well as by an academic editor with relevant expertise, and I'm writing to let you know that we would like to send your submission out for external peer review.

Once your full submission is complete, your paper will undergo a series of checks in preparation for peer review. Once your manuscript has passed the checks it will be sent out for review. To provide the metadata for your submission, please Login to Editorial Manager (https://www.editorialmanager.com/pbiology) within two working days, i.e. by May 06 2022 11:59PM.

If your manuscript has been previously reviewed at another journal, PLOS Biology is willing to work with those reviews in order to avoid re-starting the process. Submission of the previous reviews is entirely optional and our ability to use them effectively will depend on the willingness of the previous journal to confirm the content of the reports and share the reviewer identities. Please note that we reserve the right to invite additional reviewers if we consider that additional/independent reviewers are needed, although we aim to avoid this as far as possible. In our experience, working with previous reviews does save time. 

If you would like to send previous reviewer reports to us, please email me at rroberts@plos.org to let me know, including the name of the previous journal and the manuscript ID the study was given, as well as attaching a point-by-point response to reviewers that details how you have or plan to address the reviewers' concerns. 

Kind regards,

Roli Roberts

Roland Roberts

Senior Editor

PLOS Biology

rroberts@plos.org

---

## [Decision Letter · Decision Letter 1]

15 Jul 2022

Dear Rob,

Thank you for your patience while your manuscript "Comparative analysis reveals the long-term co-evolutionary history of parvoviruses and vertebrates." was peer-reviewed at PLOS Biology. It has now been evaluated by the PLOS Biology editors, an Academic Editor with relevant expertise, and by two independent reviewers. 

Based on the reviews, we will continue to consider this manuscript for publication, provided you satisfactorily address the points raised by the reviewers. I can't give a stronger commitment to publish at this moment because the Academic Editor is travelling, and they may raise additional concerns when they return; please accept my apologies for this - it seems unlikely that they will have major concerns beyond those of the reviewers, but I just thought I would be upfront. Please also make sure to address the following data and other policy-related requests.

a) Please address the concerns raised by the two reviewers.

b) Please address my Data Policy requests below; specifically, we need you to supply the numerical values (and/or, in this case, treefiles or alignments) underlying Figs 1ABCD, 2ABCDEF, 3, 4ABC, 5ACD, 6A, S4ABC, S7ABCDEFGHI, S8ABC, S9CD, S11AB, S12, S14AB, S15AB, either as a supplementary data file or as a permanent DOI’d deposition like Zenodo.

c) Please also cite the location of the data clearly in each Fig legend, e.g. “The data underlying this Figure can be found in https://doi.org/10.5281/zenodo.XXXXX" or "...can be found in S1 Data”

We expect to receive your revised manuscript within two weeks. 

*Published Peer Review History*

*Press*

Sincerely,

Roli

Roland Roberts, PhD

Senior Editor,

rroberts@plos.org,

PLOS Biology

DATA POLICY:

Regardless of the method selected, please ensure that you provide the individual numerical values that underlie the summary data displayed in the following figure panels as they are essential for readers to assess your analysis and to reproduce it: Figs 1ABCD, 2ABCDEF, 3, 4ABC, 5ACD, 6A, S4ABC, S7ABCDEFGHI, S8ABC, S9CD, S11AB, S12, S14AB, S15AB. NOTE: the numerical data provided should include all replicates AND the way in which the plotted mean and errors were derived (it should not present only the mean/average values).

DATA NOT SHOWN?

REVIEWERS' COMMENTS:

Reviewer #1:

Campbell et al. have employed multiple methods to map the diversity of endogenous parvovirus sequences in vertebrate genomes. They both expand the known diversity of these ssDNA viruses, while further dating their divergence with vertebrate evolution. Through these findings, the authors provide an evidence-based narrative for host adaptation and provide insight into host-virus coevolutionary dynamics. In particular, parvovirus phylogenies track with mammalian vicariance. The authors further provide evidence for much more extensive rodent parvovirus clades, that seem to include recombination events of VP/Cap genes - this seems like a readied reservoir of dispersal and mammalian host-switching as to give parvo success in becoming more host-ubiquitous. They have further collated their data into an open, cross-platform database (GLUE) that should be a standard of the field.

The systematic review of endogenous viral elements (EVEs) in existing genome data to reconstruct virus evolution history is entwined in an understanding of the host evolution as well. Beyond retroviral elements, identifying EVEs of other viral classes due to rare germline incorporation events seems critical in understanding the virus-host co-evolution taking place under varied transmission regimes through time.

I have a few comments and need for clarification from the authors:

Substitution models. The authors utilized GTR to construct a phylogeny of EPV ORF fragments. This makes sense in clocking variation post-genome incorporation (even while the authors acknowledge in Discussion the unknown of selection/drift on their maintenance). Do the authors see any framework under which host-specific substitution rates (often driven by selection, particularly during host shifts) could make determining the divergence of Parvo ORFs imprecise? Perhaps the large temporal scale here and match to vicariance is just too parsimonious… but I do wonder about the role of rodent host-shifts and recombinations also creating non-clock temporal abnormalities in the substitution rate that could make incorporation events primed for poor dating.

Relative frequency in mammalian genomes (Results, page 7, Table 2). Can you expand on this and/or speak to the significance? Given the paucity of genomes to capture full vertebrate diversity, and the relative over-representation of mammalian species relative to their 'weight' in biodiversity… do the authors believe this mammalian bias would hold? Are we missing a number of events in non-mammalian vertebrates that could shift your conclusion?

Page 7, line 1: *greater or less* than RT DNA viruses? A word seems to be missing.

Reviewer #2:

This manuscript by Campbell and colleagues is a comprehensive examination on the long-term evolution of parvoviruses. This study is complex but interesting and compelling as the authors catalogue EPV sequences from 752 vertebrate genomes and they do so in a manner which is highly transparent and reproducible in the form of an open database. GLUE is a computational framework that not captures sequence data and associated metadata but also allows for genomic analysis to be performed in an efficient, standardised and reproducible manner. The software itself is versatile as demonstrated by its utility to a wide range of different viruses such as SARS-CoV-2, HIV and HCV. In essence the platform to support this type of analysis is excellent. However, with that said I think that the framework and its application has been published for other viruses in the past so at times it seems that this manuscript is sometimes too focused on the novelty of Parvovirus-GLUE when the authors could have easily found over 300 EPV sequences without this framework. Overall, the manuscript is quite dense and there is a lot to unpack and while the authors attempt to navigate the reader through the study I do think that clarity is needed in places as some sections could easily be reduced or even completely removed. For example, one could easily assume that novel exogeneous virus species may be identified when doing such large data screening approaches but in the absence of other experimental support they are purely in silico findings and should be treated with caution. Also do such findings really add to the study? 

Other comments:

1. Is there a minimum sequence threshold for an endogeneous parvoviral element as I am thinking the phylogenetic signal with smaller fragments may be quite low so what quality control steps are there. 

2. Page 8 Line 11: "Fig.3a" and "Fig.3b) don't exist. Is this supposed to be Figure 3?

3. Can the authors be more explicit in how time-calibrated vertebrate phylogenies were performed. The text merely states TimeTree which I presume the authors favoured over Bayesian methods such as BEAST due to its speed? 

4. What about the limitations of sampling as I am sure it heavily biased towards certain areas and I wonder how this limits modelling the ancestral biogeographical range of protoparvovirus hosts. The authors should make reference to the limitations of this study such that it is really based on opportunistic sampling available in GenBank. 

5. Will the Parovirus-GLUE project be updated and maintained in the future?

---

## [Editor Report · Decision Letter 2]

6 Sep 2022

Dear Rob,

Thank you for your patience while we considered your revised manuscript "Comparative analysis reveals the long-term co-evolutionary history of parvoviruses and vertebrates." for publication as a Research Article at PLOS Biology. This revised version of your manuscript has been evaluated by the PLOS Biology editors and the Academic Editor.

As mentioned in my previous decision letter, we had had to proceed with the last decision without the final approval of the Academic Editor, who was travelling at the time. I've now assessed your revisions, which largely address the points previously raised; the Academic Editor shares my positive assessment, but has a few additional requests that we would like you to address before publication. Sorry that weren't able to include these in the previous round.

FINAL REQUESTS FROM THE ACADEMIC EDITOR [lightly edited]:

1) The legend to Figure 4 legend more explicitly say where the mammalian phylogeny comes from. This is said somewhere in the methods, but it is hard to find it, and it would not hurt to have this info also in the figure legend.

2) In a previous version of the manuscript, I think that the phylogenetic analysis used to have a section of its own in the main text. In my opinion it was helpful for the reader to find this information in just one place. Now this seems to be scattered around different methodological sections in the main text and some bits seem to have been lost (e.g. I could not find the reference for the GTR model of sequence evolution in the main text). I am not sure if this was done to follow the reviewer’s suggestions, but I do have to say that I like to have a place where I can go and quickly find the general overview about the phylogenetic analysis. In fact, it might be worth adding even more information on the phylogenetic analysis as it had before (apart from table 3). Given the limited time I had to read the latest version, I might have missed this in my last reading, if so sorry. On the other hand, having all this information in one place could have helped me (and others) to quickly find all the info on phylogenetic analysis.

3) I think the authors could more explicitly say (simply add this information in the text) if the phylogeographic analysis was done solely in the maximum likelihood tree (that was my impression) and perhaps add some short discussion on how the support of the nodes might affect their inferences.

I should say that the Academic Editor was rather pressed for time by their other commitments, and asks me to convey my further apologies if s/he has missed the requested information.

We expect to receive your revised manuscript within two weeks. 

*Published Peer Review History*

*Press*

Sincerely,

Roli

Roland Roberts, PhD

Senior Editor,

rroberts@plos.org,

PLOS Biology

---

## [Editor Report · Decision Letter 3]

4 Oct 2022

Dear Rob,

Thank you for the submission of your revised Research Article "Comparative analysis reveals the long-term co-evolutionary history of parvoviruses and vertebrates." for publication in PLOS Biology. On behalf of my colleagues and the Academic Editor, Tiago Quental, I'm pleased to say that we can in principle accept your manuscript for publication, provided you address any remaining formatting and reporting issues. These will be detailed in an email you should receive within 2-3 business days from our colleagues in the journal operations team; no action is required from you until then. Please note that we will not be able to formally accept your manuscript and schedule it for publication until you have completed any requested changes.

Best wishes,

Roli

Senior Editor

PLOS Biology

rroberts@plos.org